# Synergistic effects of mixing and strain in high entropy spinel oxides for oxygen evolution reaction

Jihyun Baek [1,7], Md Delowar Hossain[2,3,7], Pinaki Mukherjee[4], Junghwa Lee[5,6], Kirsten T. Winther[3], Juyoung Leem [1], Yue Jiang [1], William C. Chueh [5,6], Michal Bajdich [3] ✉ & Xiaolin Zheng [1] ✉

Developing stable and efficient electrocatalysts is vital for boosting oxygen evolution reaction (OER) rates in sustainable hydrogen production. High-entropy oxides (HEOs) consist of five or more metal cations, providing opportunities to tune their catalytic properties toward high OER efficiency. This work combines theoretical and experimental studies to scrutinize the OER activity and stability for spinel-type HEOs. Density functional theory confirms that randomly mixed metal sites show thermodynamic stability, with intermediate adsorption energies displaying wider distributions due to mixing-induced equatorial strain in active metal-oxygen bonds. The rapid sol-flame method is employed to synthesize HEO, comprising five 3d-transition metal cations, which exhibits superior OER activity and durability under alkaline conditions, outperforming lower-entropy oxides, even with partial surface oxidations. The study highlights that the enhanced activity of HEO is primarily attributed to the mixing of multiple elements, leading to strain effects near the active site, as well as surface composition and coverage.

The transition from fossil fuel-based to renewable energy-based society requires efficient and clean energy carriers, such as electrons and hydrogen[1,2]. Electrochemical water splitting, i.e., water electrolysis, converts electricity to hydrogen and oxygen[3,4], but its efficiency is hampered by the slow kinetics of oxygen evolution reaction (OER) at the anode, which calls for efficient and stable electrocatalysts to reduce the reaction barrier for OER[5]. To date, the most active and stable catalysts for OER are noble metal-based oxides, such as $IrO_2$ and $RuO_2$[6–10]. However, the scarcity and high cost of Ir and Ru have limited the scalable adoption of water electrolysis, and the search for noble-metal-free OER electrocatalysts is being actively pursued. Many strategies have been suggested to improve OER activity for non-noble metal-based electrocatalysts[11], including the modification of chemical composition[12], nano/microstructure[13], phase distribution[14], or alloying[15]. Among those non-noble metal-based catalysts, the 3d transition metal oxides with different phases (rock-salt, perovskite, or spinel) and (oxy)hydroxide show low overpotential at the desired current densities (>10 mA cm$^{-2}$) in alkaline electrolyte[16–23].

A new class of material, i.e., high-entropy materials (HEMs), such as high-entropy oxides (HEOs), has emerged as a new class of electrocatalysts. HEMs, consisting of five or more equimolar cations in a single phase[24], have already attracted great interest in energy storage[25,26], fuel cell[27,28], and catalysis[15,29,30], due to their excellent tunability from mixing different species and the concentration of constituents[31–35] as well as the physical and chemical stability from the large entropy of mixing[35,36]. HEOs are particularly of interest as OER

[1]Department of Mechanical Engineering, Stanford University, Stanford, CA 94305, USA. [2]SUNCAT Center for Interface Science and Catalysis, Department of Chemical Engineering, Stanford University, Stanford, CA 94305, USA. [3]SUNCAT Center for Interface Science and Catalysis, SLAC National Accelerator Laboratory, Menlo Park, CA 94025, USA. [4]Stanford Nano Shared Facilities, Stanford University, Stanford, CA 94305, USA. [5]Department of Materials Science and Engineering, Stanford University, Stanford, CA 94305, USA. [6]Stanford Institute for Materials and Energy Science, SLAC National Accelerator Laboratory, Menlo Park, CA, USA. [7]These authors contributed equally: Jihyun Baek, Md Delowar Hossain. ✉e-mail: bajdich@slac.stanford.edu; xlzheng@stanford.edu

electrocatalysts. Indeed, different crystal structures of HEOs, such as rock-salt[37,38], perovskite ($ABO_3$)[39], spinel-type ($AB_2O_4$)[40,41], and (oxy) hydroxide (MOOH)[42] have been evaluated as OER catalysts. Zhang et al. reported that the entropy-engineered rock-salt $(CoNiMnZnFe)_3O_{3.2}$ shows an overpotential of 336 mV at 10 mA cm$^{-2}$ and good stability (retained 86.9% of current density for 20 h) for OER[37]. Nguyen et al. studied La-based perovskite HEO with different B site cations (Cr, Mn, Fe, Co, and Ni) for OER. It shows an overpotential of 325 mV at 10 mA cm$^{-2}$ and stability for 50 h without apparent degradation[39]. Zhang et al. investigated spinel-based HEO $(Co_{0.2}Mn_{0.2}Ni_{0.2}Fe_{0.2}Zn_{0.2})Fe_2O_4$ for OER, exhibiting an overpotential of 326 mV at 10 mA cm$^{-2}$ and maintaining 89.4% of current density after 10 h[41]. In addition to the experimental efforts, various computational techniques have been utilized to understand the thermodynamic stability and surface catalytic processes for HEOs[43]. In the recent computational work, Anand et al. optimized the lattice configuration of cations and anions to examine how the elemental composition of HEO, (NiMgCoCuZn)O, affected enthalpy and configurational entropy[44]. More recently, Rossmeisl et al. computationally predicted oxygen reduction reaction (ORR) electrochemical activity via calculating O* and OH* binding energy distributions for different active sites of HEAs[45]. By the use of DFT adsorption energies prediction, Svane et al. theoretically optimized the composition of rutile HEO (110) surface based on Ir, Ru, Ti, Os, and Rh for OER[46]. These pioneering studies suggest that randomly mixed cations modulate their valence states, electronic structures, and adsorption energies of the intermediate species, e.g., *O, *OH, and *OOH, offering the possibility to tune and improve their catalytic properties towards OER[12,46,47]. However, a thorough investigation of the catalytic properties of HEOs for OER is still deficient both experimentally and theoretically due to the complication involved in the synthesis and the lack of a proper computational model to describe stability and OER activity. Moreover, identifying active sites for OER in HEOs and their activity origin is still unrevealed.

Here, we study spinel structure of HEOs for OER as they have tunable physicochemical properties from different cation arrangements in the tetrahedral and octahedral sites, which affects catalytic properties. To examine the OER activity and stability for spinel HEOs comprising Co, Fe, Ni, Cr, and Mn, we first computationally develop an HEO impurity model. Then, we evaluate the HEO stability for various active sites by calculating mixing entropies and enthalpies of a homogenous mixture. Guided by the theoretical results, we experimentally synthesize spinel HEO (Co, Fe, Ni, Cr, and Mn) by the sol-flame method[48,49]. The morphological and structural properties of the synthesized HEO are characterized via X-ray diffraction (XRD), scanning transmission electron microscopy (STEM), and energy-dispersive X-ray spectroscopy (EDS) elemental mapping. X-ray photoelectron spectroscopy (XPS), X-ray absorption spectroscopy (XAS), and electron energy loss spectroscopy (EELS) are further used to investigate the surface and bulk oxidation states of the HEO constituents. Moreover, we evaluate and compare the OER performance for HEO to those of the lower entropy oxides ranging from binary to quinary in alkaline electrolyte. Further, we use our HEO impurity model to predict OER activity for various metal active sites via calculating O* and OH* adsorption energies and overpotentials and their connection to the contraction and expansion of metal-oxide bonds. Finally, we discuss overall OER activity trends based on the electrochemical characterizations..

## Results

### HEO impurity model

The initial studies of unary, binary, and ternary mixtures of Cr, Mn, Fe, Co, and Ni are described in the Methods Section. The stability for various binary and ternary spinel oxides is computed (Supplementary Figs. 1–3) by using DFT calculation and Crystal Field Theory. In the case of more complex metal oxides, rather than constructing a fully periodic bulk model of an arbitrary mixture, it is more computationally feasible to study the HEO effect via local mixing near the active site. As schematically illustrated in Fig. 1a, we select a central metal site and allow neighboring metals (or ligands) to be interchanged. Such an approach can be extended to the next-neighbor shells and beyond while also being site-specific and computationally tractable. This approach can be used to develop both bulk and surface impurity models. The surface model is applied to extract the mixing enthalpy and study the catalytic properties, such as OER, as discussed later.

For the equimolar HEO surface systems, we purposely choose $Co_3O_4(N)$ as it forms a stable normal spinel structure with high cubic

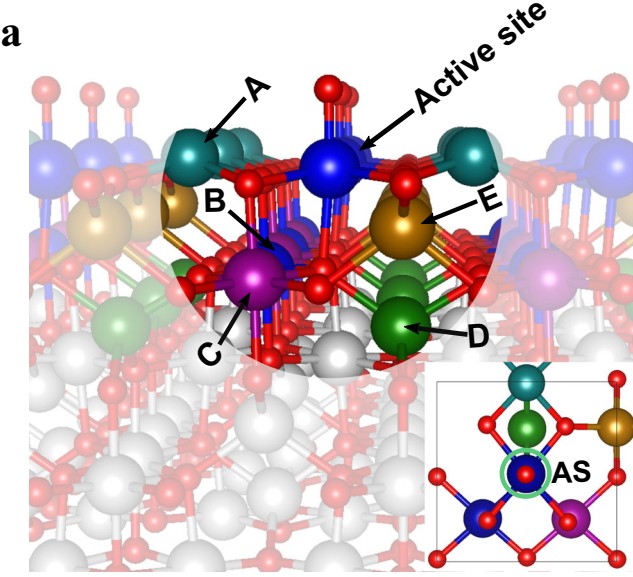

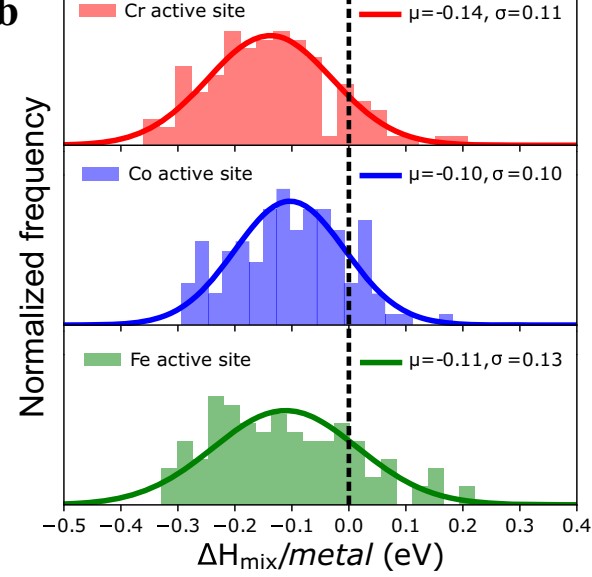

**Fig. 1 | HEO impurity model for high entropy spinel $M_3O_4$. a** The HEO system containing the OER active site is shown in the center with adsorbed O* (red) surrounded by 5 metal neighbors (labeled as A to E for Cr,Co,Mn,Ni,Fe) within a bond length of 3.5 Å. Inset figure shows the top view of HEO containing active sites (AS) surrounded by neighbors atoms. **b** Relative mixing enthalpy per mixing metals for various active sites of the HEO surface as referred to their bulk systems.

symmetry (Supplementary Table 1). Arguably, some other systems or volume-averaged spinel models could also have been used. Here, we choose the (100) facet terminated with the octahedral site (100)-A type surface of $Co_3O_4$ from the literature[50] (see Methods Section). In this setting, each active site at the top-most layer is surrounded by the five nearest elements (neighbors) (<3.5 Å) (A-E, Fig. 1a), and these five neighboring sites are assumed to be occupied by exactly one atom of each of the elements Cr, Co, Mn, Ni, and Fe in any arrangement, which leads to 120 distinct permutations per active site. The location of five elements is important, which affects active site activity significantly. Similar to Rossmeisl et al. approach for high entropy alloys[45], this strategy allows the local and tractable treatment of the HEO catalytic effects at the surface. Including the next-neighbor shells (within 6.6 Å) in our model would result in a 100-fold increase of distinct permutations. However, based on a few cases, we determined that this effect is less than 0.3 eV per adsorption energy. Moreover, we limit our study to Co, Cr, and Fe as active sites because both Co and Cr spinel oxides show the best activity in pure form for OER, while Fe active sites represent weak binding limits (Fe, Mn, Ni) (discussed later). The likelihood that Fe-site behaves similarly in Fe-doped NiOOH[51] and related HEO systems[12] is also a relevant factor.

## Stability of HEO

The feasibility of successful HEO formation is estimated thermodynamically by calculating the Gibbs free energy of mixing ($\Delta G_{mix}$) using Eq. (1) as a more negative value of $\Delta G_{mix}$ indicates a more stable and homogenous mixing.

$$\Delta G_{mix} = \Delta H_{mix} - T\Delta S_{mix} \tag{1}$$

We evaluated the theoretical configurational entropy of mixing $\Delta S_{mix}$ for an equimolar 5-metal system by using the following formula as

$$\Delta S_{mix} = -R \sum_{i=1}^{M} x_i \ln(x_i) \tag{2}$$

where R is the ideal gas constant (0.0000862 eV K$^{-1}$ metal$^{-1}$); M is the number of metal cations in HEOs; $x_i$ is the molar fraction of each metal cation. For a 5-component (M = 5) equimolar system, $\Delta S_{mix} = 1.6R$ and the calculated configuration mixing entropy at the flame annealing temperature of 1000 K is $-T\Delta S_{mix} \sim -0.14$ eV/metal. We used our surface HEO impurity model to evaluate the enthalpy of mixing, $\Delta H_{mix}$ per metal referenced to pure $Co_3O_4$ (100) surface simply as

$$\Delta H_{mix} = \left[ E_{HEO(100)}^{slab} - E_{Co_3O_4(100)}^{slab} - \sum_{i=1}^{M} E_i^{bulk}\left(M_3^i O_4\right) + M \cdot E^{bulk}(Co_3O_4) \right] / M \tag{3}$$

where $E_{HEO(100)}^{slab}$ and $E_{Co_3O_4(100)}^{slab}$ are the DFT total energies for the HEO and $Co_3O_4$ (100) slabs, respectively. In Eq. (3), the spinel bulk energies for individual mixing metals (M=Co, Fe, Ni, Cr, Mn) are labeled as $E_i^{bulk}\left(M_3^i O_4\right)$. Because surface energies of the HEO (100) model and $Co_3O_4$ (100) approximately cancel out, the mixing enthalpy of this surface model is also a reasonable estimation of the bulk HEO mixing.

Fig. 1b plots the distribution of the resulting mixing enthalpies for the Co, Cr, and Fe active sites. The distribution of $\Delta H_{mix}$ per metal for all three active sites is rather symmetric around the mean value and spans up to 0.5 eV, which is much larger than that of the rated HEA study (-0.06 eV)[52] by considering an equimolar five elements system. Moreover, over 85% of combinations have a negative $\Delta H_{mix}$. Since the $-T\Delta S_{mix}$ is also negative (−0.14 eV/metal), $\Delta G_{mix}$ will be mostly negative, indicating the formation of HEOs is thermodynamically favorable.

## Synthesis and characterization of HEOs

Fig. 2a illustrates the sol-flame process[48,49] for synthesizing spinel HEO (Co, Fe, Ni, Cr, and Mn) as described in the Methods section. It entails precursor mixing in the liquid phase, followed by spin-coating, rapid high-temperature flame annealing and quenching. The high-temperature annealing and subsequent quenching process are crucial to preventing the formation of intermetallic phases[15,53]. The X-ray diffraction (XRD) of the resulting HEO in Fig. 2b shows a pure spinel Fd-3m (227) crystal structure with ICDD 00-010-0325. The in-situ temperature-dependent XRD results (Supplementary Fig. 4) show that the spinel structure starts to appear at 400 °C. In-situ temperature-dependent XRD is operated by a slow heating process, which is different from the flame annealing process including fast heating and cooling. We can still get insight into how the HEO crystal structure is developed at different temperatures. Thus, the spinel structure is prominent at the flame treatment temperature of 1000 °C, indicating that the sol-flame temperature is high enough to synthesize the spinel structure of HEO. The tiny diffraction peak at 33° represents alpha-$Fe_2O_3$ probably formed due to the presence of excess iron precursors in the mixture solution, which could not participate in the formation of HEO[54–56]. Moreover, the transmission electron microscopy (TEM) (Fig. 2c) and scanning electron microscopy (SEM) (Supplementary Fig. 5) images show that the as-synthesized HEO consists of interconnected nanoparticles (tens of nm in diameter) that are uniformly coated on the FTO/glass substrate. In addition, the high-resolution TEM image (Fig. 2d) and the selected area electron diffraction (SAED) pattern (Fig. 2e) reveals that those nanoparticles are the polycrystalline phase of spinel oxide, mainly indexed to (220), (311), and (222), corresponding to the d-spacings of 0.290, 0.252, and 0.240 nm, respectively. The d-spacing values slightly deviate from the reference $NiFe_2O_4$ (00-010-0325), 0.2948, 0.2513, and 0.2408 nm for (220), (311), and (222), which is probably attributed to the atoms with different ionic radius in the lattice. The fast Fourier transform (FFT) image (Fig. 2d, inset) further confirms that the lattice fringes agree well with the XRD and scanning transmission electron microscopy (STEM) results. Fig. 2f shows the dark-field STEM and energy dispersive X-ray spectroscopy (EDS) elemental mapping images, verifying the homogeneous mixing of each element (Co, Fe, Ni, Cr, Mn, and O).

X-ray photoelectron spectroscopy (XPS) (Supplementary Fig. 6) was employed to investigate the compositional and chemical structure at the surface[39]. The quantification analysis shows that the HEO surface has a near-equimolar concentration of each metal cation. Detailed elemental XPS analysis (Supplementary Fig. 7) shows that Cr exists as $Cr^{3+}$ and other elements (Mn, Fe, Co, and Ni) have both divalent and trivalent oxidation states. The O 1$s$ spectra were also deconvoluted to three different peaks; M-O at 529.8 eV, M-OH at 531.2 eV, and adsorbed $H_2O$ at 532.5 eV, suggesting that the synthesized HEO particle surface contains hydroxide or oxyhydroxide species as well. For binary oxides to HEO, we deconvoluted the detailed Fe 2$p_{3/2}$ and Co 2$p_{3/2}$ XPS spectra to see the different ratios of divalent and trivalent oxidation states, as shown in Supplementary Fig. 8. They exhibit different $Fe^{2+}/Fe^{3+}$ and $Co^{2+}/Co^{3+}$ at the surface where HEO shows the higher $Fe^{2+}/Fe^{3+}$ and lower $Co^{2+}/Co^{3+}$ than the lower entropy oxides. This implies that HEO has more $Fe^{2+}$ and $Co^{3+}$ at the surface, elaborated by electron energy loss spectroscopy (EELS) in the later section. Thus, the different ratios of divalent and trivalent states in Co and Fe 2$p_{3/2}$ influence the OER activity. Also, the average concentrations of trivalent and divalent states of all elements for different spinel oxides are summarized in Supplementary Table 2. Based on the values in Supplementary Table 2, we summarized the $Co^{3+}/Co^{2+}$ values for the different spinel oxides (Supplementary Fig. 9) since $Co^{3+}$ was found to be one of the active sites as shown in Fig. 1.

Furthermore, the X-ray absorption spectroscopy (XAS) (Supplementary Fig. 10) shows that the bulk oxidation states of Co and Fe in

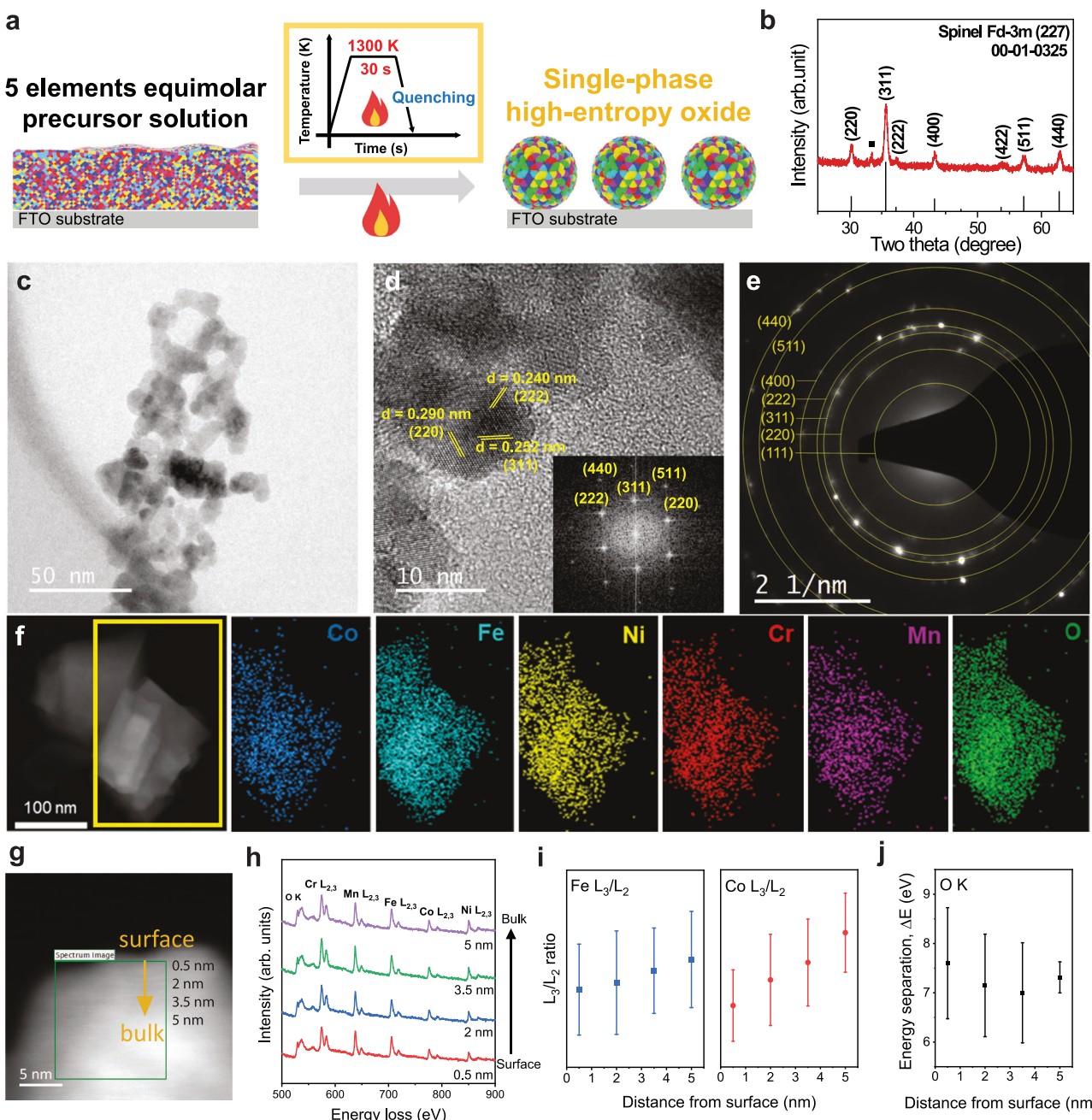

**Fig. 2 | Characterizations for as-synthesized spinel HEO. a** Schematic of the sol-flame synthesis process. **b** XRD and **c, d** high-resolution TEM images of HEO (inset: FFT image). **e** SAED pattern and **f** EDS-mapping images of Co, Fe, Ni, Cr, Mn, and O. **g** Dark-field image of HEO particles and **h** EEL spectra extracted from the four points with a distance of 1.5 nm between each point (0.5 ~ 5 nm) in (**g**), showing O K-edge and L2,3-edges of Co, Fe, Ni, Cr, and Mn. **i** Fe L3/L2 and Co L3/L2 white-line intensity ratio for the four points (0.5 ~ 5 nm) in (**g**). **j** The energy difference between O K-edge pre-peak and main peak (energy separation, ΔE) at the four points (0.5 ~ 5 nm) in **g**.

HEO contain both divalent and trivalent states. Based on the X-ray absorption near-edge structure (XANES) profiles, the quantitative comparison of oxidation state from the Co K-edge spectra (Supplementary Fig. 10a, b) shows that the valence states of Co for both CoFeNiCr and HEO are located between $Co^{2+}$ and $Co^{2+}/Co^{3+}$. In addition, the higher Co valence state appears with adding Mn cation to CoFeNiCr, which is consistent with the XPS result. Moreover, the Fe K-edge of CoFeNiCr and HEO is shown in Supplementary Fig. 10c, and the Fe oxidation state (Supplementary Fig. 10d) is in the lower region than the reference materials, $Fe^{2+}/Fe^{3+}$ and $Fe^{3+}$ in which the relative Fe oxidation state of HEO is the lowest and tends to appear in a more divalent state. Thus, HEO exhibits relatively higher $Co^{3+}$ than $Co^{2+}$ and

$Fe^{2+}$ than $Fe^{3+}$ throughout the particles, compared to those of the lower entropy oxide (CoFeNiCr).

The electron energy loss spectroscopy (EELS) analysis (Fig. 2g–j) was performed on the HEO particle to quantify its chemical composition. The EEL spectra (Fig. 2h) show the presence of the $L_{2,3}$-edges of all five metal elements (Cr, Mn, Fe, Co, Ni) at four positions in Fig. 2g with a distance of 1.5 nm. Moreover, Fig. 2i plots the $L_3/L_2$ white-line intensity ratio for Co and Fe at those six positions. As the $L_3/L_2$ white-line intensity ratio was reported to be closely associated with the orbital occupancy of the 3d-transition metal elements and their oxidation states[57–60], the results suggest that the HEO has more $Fe^{2+}$ and $Co^{3+}$ on the surface and more $Fe^{3+}$ and $Co^{2+}$ toward the center of the

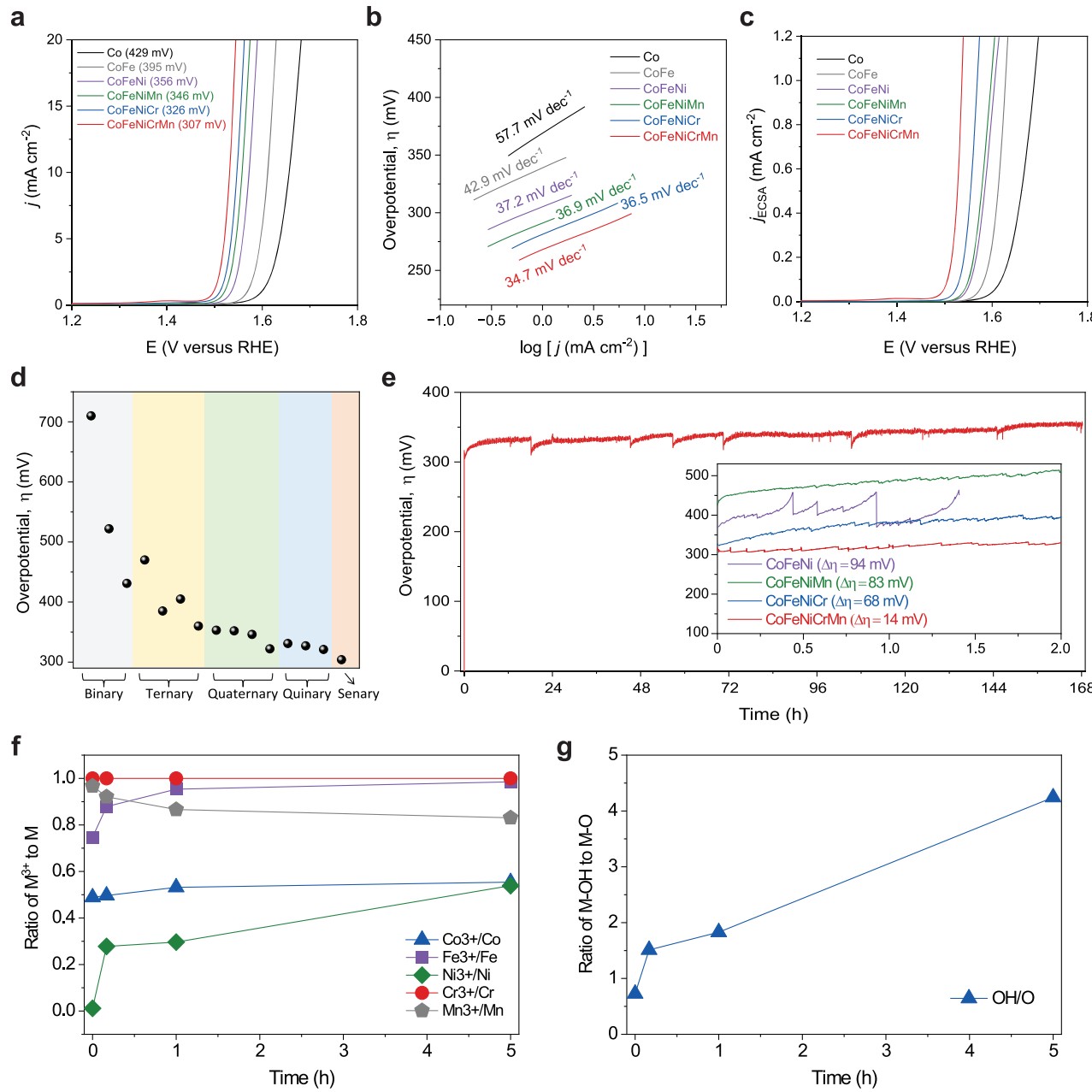

**Fig. 3 | Electrochemical response of various spinel oxides. a** Linear sweep voltammetry (LSV) normalized to the geometric area with 90% iR correction in 1 M KOH electrolyte and (**b**) their Tafel plots. **c** Normalized LSV plots to the ECSA. **d** Overpotential plot as a function of the number of cations in oxides. **e** Chronopotentiometry (CP) curves of HEO anode (inset: CP comparison with other control groups) **f, g** Ex-situ XPS analysis. **f** the overpotential change and ex-situ XPS quantification results of the ratio of $M^{3+}$ (M = Co, Fe, Ni, Cr, and Mn) to M over time. **g** The ratio of OH to (OH + O) over time.

HEO particle. Here, the error bars come from several spots of the different particles, and each value is shown in Supplementary Fig. 11. Fig. 2j shows the energy separation between O K-edge pre-peak to the main peak, and its larger values near the surface indicate the higher oxidation states of overall elements on the surface in which HEO has the random energy separation value at the four positions[61]. This is probably from the different oxidation states of five elements mixed from the surface to the bulk.

### Electrochemical performance of high-entropy spinel oxides
Comparisons of the polarization curves of different spinel oxides containing one to five metal cations in a 1 M KOH electrolyte are shown in Fig. 3a. Among them, the HEO (CoFeNiCrMn) has the lowest average overpotential of 307 mV at 10 mA cm$^{-2}$, in comparison to the other

spinel oxides: CoFeNiCr (326 mV) < CoFeNiMn (346 mV) < CoFeNi (356 mV) < CoFe (395 mV) < Co (429 mV), and <Fe (522 mV) (Supplementary Fig. 12) with 90% iR correction. The different overpotential values in terms of different iR correction percentages are also shown in Supplementary Fig. 13 and Supplementary Table 3. To assess the reproducibility of the same composition, we conducted electrochemical measurements for at least three samples of the same composition and calculated the error bars. The slight variations in current densities among samples are attributed to experimental factors, such as substrate resistance, surface coverage, and other relevant variables. Starting from Co, adding Fe is slightly beneficial for activity improvement, and adding Ni to CoFe hugely lowers the overpotential. Combining Mn to CoFeNi slightly improves the performance but shows large fluctuation (sometimes worse than CoFeNi). Interestingly, adding

Cr is hugely beneficial for the activity. Finally, HEO with all five elements shows the highest performance, which probably stems from the randomly mixed elements and tuned adsorption energies discussed later. The HEO also has the smallest Tafel slope (34.7 mV dec$^{-1}$) than other spinel oxides (Fig. 3b). In addition, the electrochemical impedance spectroscopy (EIS) measurement shows that HEO has the lowest charge transfer resistance at the interface between the catalyst and electrolyte (See details about the equivalent circuit model and element description in Supplementary Fig. 14 and Supplementary Table 4)[62]. It suggests that HEO has higher electrical conductivity than the lower entropy oxides. Furthermore, utilizing the measured ECSA along with the corresponding double-layer capacitance, Fig. 3c illustrates the polarization curves of various spinel oxides normalized by ECSA. A comprehensive description of the ECSA can be found in Supplementary Fig. 15. The roughness factor was also determined by dividing the ECSA by the geometric surface area of the electrode. For the respective materials FTO, Co, CoFe, CoFeNi, CoFeNiMn, CoFeNiCr, and HEO, the resulting roughness factor values are 1.2, 15.2, 20.4, 41.6, 46.9, 20.0, and 23.3, respectively. The LSV curve trend after normalizing by ECSA depicted in Fig. 3c shows the same as those normalized by the geometric area. However, the normalized current density appears to decrease due to the significantly higher electrochemical surface area. For HEOs, we also fixed the four metal cations of CoFeNiMn and varied the fifth one (Cr, V, Cu, Zn, and W) (Supplementary Fig. 16). Among them, the inclusion of Cr exhibits the lowest onset potential, which is consistent with the report that Cr (oxide or oxyhydroxide) is active for OER[63]. Fig. 3d summarizes the activity trend for all the overpotential values of the different spinel oxides, showing that the OER activity is improved with increasing the number of metal cations. Also, the HEO displays nearly 95% of Faradaic efficiency (FE) at 1.54 V vs. RHE, demonstrating great selectivity (Supplementary Fig. 17).

The stability of HEO was further evaluated by chronopotentiometry and cyclic voltammetry tests. The chronopotentiometry measurement under 10 mA cm$^{-2}$ shows that the overpotential for HEO increases by ~12% for 168 h (7 days) (Fig. 3e). It should be noted that part of the change is related to surface coverage by generated oxygen bubbles that were not entirely removed during the measurement. In comparison, spinel oxides with three and four metal cations show a pronounced 27% and 22% increase in overpotentials within 2 h (Fig. 3e, inset). The trend demonstrates that stability is improved with a larger degree of mixing. Furthermore, the cyclic voltammetry test of HEO shows that the oxidation peak at approximately 1.4 V vs. RHE gradually increases during the 1000 cycles (scan rate of 100 mV s$^{-1}$) (Supplementary Fig. 18), suggesting that the surface oxidation states were altered during the OER. The OER activity and stability values for different HEMs, binary spinel oxides, and binary layered double hydroxides (LDH) from other literature are also summarized for comparison in Supplementary Table 5. Here, we conducted the durability test for the longest time among the catalysts in the table, and HEO shows comparable stability and activity compared to the reported materials, even though we simply synthesized it as a thin film, not including complex synthesis and engineering. In addition, time-dependent ex-situ XPS analysis was performed to analyze the surface state change of HEO. Fig. 3f shows that the ratio of the trivalent state in Co, Fe, and Ni and the divalent state in Mn increase within a couple of hours of the test. The O 1s XPS spectra in Fig. 3g show that the surface OH out of (OH + O) became dominant with time. Detailed XPS analysis in Supplementary Fig. 19 indicates that the Co$^{3+}$, Fe$^{3+}$, Ni$^{3+}$, and OH concentrations were increased at the surface for 5 h, mainly in Fe$^{3+}$ and Ni$^{3+}$, in which this result demonstrates that the surface (M)OOH (M = Co, Fe, Ni, and/or Cr) was formed during the electrochemical measurement, which influenced the slight change of overpotential. Although it is well known that the surface oxyhydroxide is also a promising OER catalyst[63], HEO still plays an important role as support contributing to the great activity. A previous report shows

570 mV and 500 mV of the overpotential for FeOOH and NiFeOOH at 2.5 mA cm$^{-2}$, respectively[64]. Even other oxyhydroxides including NiFe, CoFe, CoCr, FeCr, and CoFeCr all display higher overpotential than our HEO[65,66]. In addition, we evaluated the surface oxidation OER activity on oxyhydroxide by the DFT calculation that will be elaborated on in the later section. In addition, we investigated how much the constituent elements were dissolved to the electrolyte over time using inductively coupled plasma mass spectrometry (ICP-MS) in Supplementary Fig. 20. We found that mostly certain amount of Cr was dissolved, which is recognized from the previous report[67], based on the certificate of analysis for the trace elements in water from the National Institute of Standards & Technology (NIST)[68].

## Theoretical investigation of OER activity

After experimental synthesis and OER activity/stability measurements, we found that the spinel HEO shows lower overpotential than lower entropy oxides. To interpret the experimental activity, we calculated the O* and OH* binding energies for three different active sites (Cr, Co, and Fe) using the HEO impurity model (see Fig. 1), along with their pure spinel activity. The results are summarized in a 2D volcano (O*-OH*, OH* descriptors) overpotential heat map (Fig. 4a), developed previously[10,47,63,69] and based on the scaling in Supplementary Fig. 21. By considering only the octahedral (+3) active sites at the (100) facets, our calculations show that the Co$_3$O$_4$ spinel has the smallest overpotential of 0.63 V among all pure spinel oxides. The trends of overpotential for various pure spinel oxides are Co$_3$O$_4$ (0.63 V) < Cr$_3$O$_4$ (0.64 V) < Mn$_3$O$_4$ (0.91 V) < Ni$_3$O$_4$ (1.25 V) < Fe$_3$O$_4$ (1.76 V). The binding energy for all adsorbates and their corresponding overpotentials are tabulated in Supplementary Table 6 and plotted in Fig. 4a. Next, we introduce the O* and OH* binding energies from the HEO impurity model (see Fig. 1) for Cr, Co, and Fe active sites. We predicted more than one thousand binding energies and their corresponding overpotentials were grouped for each active site (Fig. 4a). The distributions of overpotentials (and their corresponding descriptor values) for Co (light blue) and Cr (orange) sites in HEO are densely packed, while Fe (light green) sites are widespread. The arrows represent the changes in the activity trends compared to their respective pure spinel overpotentials. In all three tested active metal types, $\Delta G_{O*}$ - $\Delta G_{OH*}$ OER descriptor values are moved from the right to the left side towards the region of stronger adsorption, and Cr shows the smallest while Fe shows the largest shift.

The overall OER activity for HEO depends strongly on the active sites. The Co-active site in HEO shows a minimum overpotential of 0.29 V, while both Cr and Fe result in a minimum overpotential of 0.34 V for the OER activity. Fig. 4b shows the OER activity distribution as a function of overpotential, extracted from the 2D volcano plot (Fig. 4a). It shows that less than 5% of the active sites are activated below 0.4 V applied overpotential and the majority contribution to activity comes from the Co-active sites. Nearly 25% of the sites are active when the overpotential is less than 0.5 V, and Co sites continue to dominate in terms of activity. Next, several Cr active sites were activated within the region of intermediate overpotential of 0.5–0.8 V. More Cr active sites are activated as the overpotential increases. Unlike Co and Cr, only a few Fe active sites are found below 0.7 V overpotential. We also compared the OER activity on the Fe-NiOOH oxyhydroxide system calculated previous[51] to the spinel systems. The theoretical activity Fe-NiOOH (0.37 V) is slightly lower than the most active Fe site in HEO (0.34 V) and the overall active Co site in HEO (0.29 V). Based on the predicted overpotentials, the majority of the overall activity is possibly contributed by the Co site, with minor contributions coming from the Cr, and Fe sites in HEO, and oxyhydroxide.

To analyze the activity trend results obtained in the experiment, we compared the OER activity of HEO with that of other spinel oxides, such as ternary and quaternary with various elemental combinations,

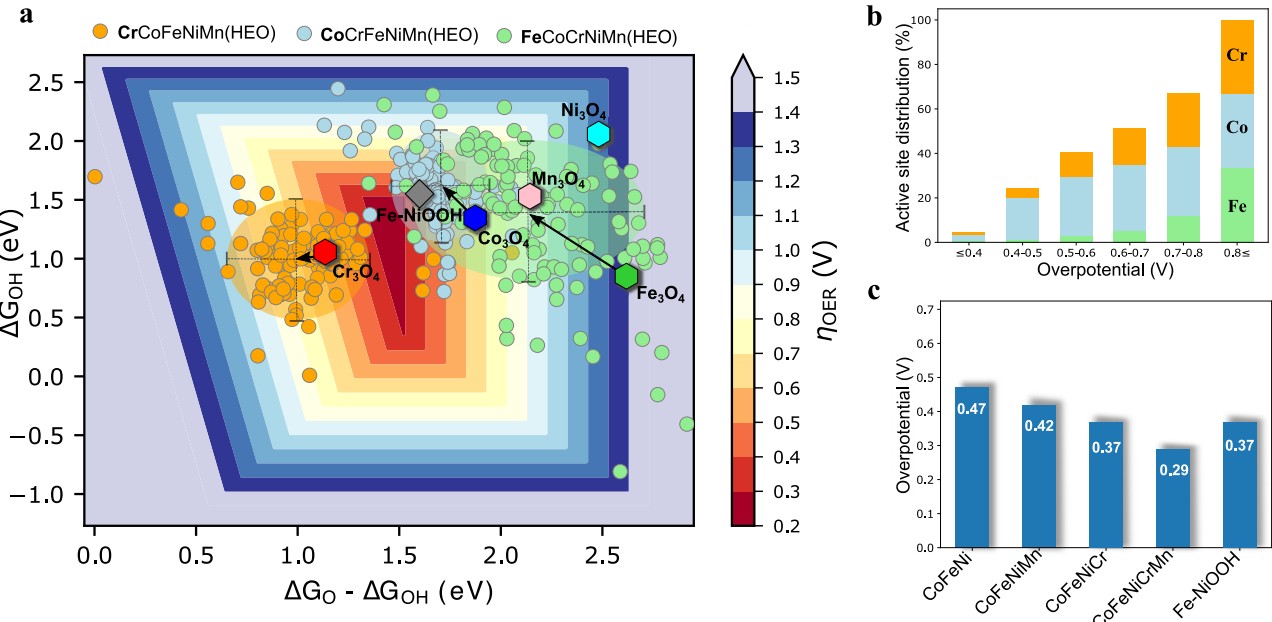

**Fig. 4 | OER site-dependent activity of the equimolar HEO. a** OER activity volcano plot as a 2D heat map of overpotentials based on binding energy calculation of O* and OH*, and scaled OOH* values for HEO systems, grouped by the active site along with the pure spinel systems. **b** Active sites (Cr, Co, Fe) distribution as a function of OER overpotential for HEO systems. **c** Predicted activity trends for different spinel oxide systems with explicitly calculated OOH*.

along with Fe-NiOOH (see Fig. 4c). We predicted that the combination of 5 elements in HEO has the lowest calculated overpotential of 0.29 V. Later we calculated overpotential by considering permutation theory for lower entropy oxides as well (Supplementary Figs. 22–24). The calculated overpotential trends for these oxide systems are as follows: HEO (CoFeNiCrMn) (0.29 V) < CoFeNiCr (0.37 V) = Fe-NiOOH (0.37 V) < CoFeNiMn (0.42 V) < CoFeNi (0.47 V). This trend shows good overall consistency with experimental findings, justifying the applicability of the HEO impurity model. Finally, we also analyzed the origin of board adsorption energy distributions for O* and OH* descriptors, as discussed in detail next.

## The role of local strain on HEO activities

The HEO $\Delta G_{O*}$ - $\Delta G_{OH*}$ OER descriptor values are generally shifted towards the region of stronger adsorption (Fig. 4a and Supplementary Fig. 25). However, the individual distributions of O* and OH* binding energies are more complex. The Cr active sites move to the left (stronger ads.), while Co and Fe move to the right (weaker ads.) (Supplementary Fig. 26). We attempted to investigate the possible reason for these changes considering two limiting cases having either strong or weak adsorption energies relative to the pure system using the Co-active site (Supplementary Figs. 27, 28). The major finding here is that the local expansion (contraction) to equatorial Co-O bonds causes the stronger (weaker) O* and OH* binding. The presence of Fe as a surface neighbor strongly attracts oxygen, causing an expansion of these lateral Co-O bonds. In contrast, when the surface neighbor is Mn, and the sub-surface neighbor is Cr, these Co-O bonds contract, resulting in weaker O* and OH* adsorption. The charge density difference analysis clearly shows that such expansion of Co-O bonds creates electron depletion along Co sites causing strong O* and OH* adsorption compared to pure and contraction cases. Finally, the equatorial expansion-contraction is also compensated by the axial sub-surface oxygen M-O bond to shorten-elongate, respectively (Supplementary Figs. 29, 30). In order to provide experimental evidence of the microstrain, we additionally measured XRD for some of the spinel oxides including HEO, and employed the XRD patterns to obtain the Williamson-Hall plot, which can be used to calculate the microstrain

and crystallite size of each spinel oxide. As shown in Supplementary Figs. 31, 32 and Supplementary Table 7, HEO shows the highest microstrain among the various spinel oxides in correlation with the higher activity. We assume that, fundamentally, the lattice structure of the multi-element solid solution phase (of HEO) is distorted due to a variety of elemental bonds, which leads to microstrain and is in good agreement with the calculation of the bonding and prediction of HEO activity. Indeed, microstrain in material structures is recognized as a significant factor impacting OER activity[70,71]. Nevertheless, we recognize that the OER performance is also influenced by other factors, such as surface composition. Therefore, it is vital to consider that microstrain is one of these factors affecting OER activity.

In summary, we developed a computationally feasible HEO impurity model to describe the stability and OER activity of any metal-oxide HEO. The model can be easily formed by local mixing near the active site instead of constructing a completely periodic bulk model of an arbitrary mixture. The mixing enthalpy calculations using this model demonstrate that the spinel HEO (CoFeNiCrMn) system is thermodynamically stable at 1000 °C. We further experimentally synthesized HEO contains five cations and other spinel oxides with one to four metal cations at 1000 °C using the sol-flame method. Our electrochemical characterization shows that HEO has the lowest overpotential of ~307 mV to reach 10 mA cm⁻², the smallest Tafel slope (34.7 mV dec⁻¹), and the best stability (~12% increment of the overpotential over 168 hrs) compared to oxides with a smaller number of metal cations. The interaction among metal cations for OER activity is rather complex. Starting from the Co binary oxide, adding Fe was negligible for the activity improvement, while Ni significantly enhanced the activity. For the quinary oxides, combining Mn with CoFeNi slightly improved the performance, while Cr was hugely beneficial. Moreover, our model also predicts that HEO has the highest activity due to the tunable adsorption energies with various active sites, especially Co, followed by Cr and Fe as well as oxyhydroxide formed during the measurement. Finally, the high OER activity in HEO originated from the broader OER activity distributions due to a large number of stable local bonding configurations. The mean values of these distributions are controlled by the local strain in the active metal

site-oxygen bonds, where equatorial contraction (expansion) leads to weaker (stronger) OER surface thermodynamics. This study demonstrates the promise of HEO as a electrocatalyst with tunable properties due to the abundance of stable thermodynamic configurations and local strain effects that result in a wide range of active sites.

## Methods

### Computational details

Spin-polarized density functional theory calculations with plane-wave basis set were performed using the Vienna ab initio simulation package (VASP)[52,72,73] and project augmented wave (PAW) pseudopotential[74]. All the calculations were conducted and analyzed using the atomic simulation environment (ASE)[75]. The Perdew-Burke-Ernzerhof functional was used to describe electronic exchange and correlation[76]. For all calculations, we applied the DFT + $U$ method and set the energy cutoff to be 500 eV[77]. The spin-polarized bulk structures of $Co_3O_4$, $Fe_3O_4$, $Ni_3O_4$, $Cr_3O_4$, and $Mn_3O_4$ were optimized using $5 \times 5 \times 5$ k-point mesh sampling[78], and their bulk structures were considered to be optimal when the energies and forces drop below $10^{-6}$ eV and 0.01 eV/Å, respectively. We applied initial magnetic moments to achieve targeted spin states on each element by setting the occupation matrix of $3d$-electron localization, followed by geometry optimizations[79].

### Calculation of bulk spinel oxides

We calculated the bulk spinel structures of $Co_3O_4$, $Fe_3O_4$, $Ni_3O_4$, $Cr_3O_4$, and $Mn_3O_4$ using a Hubbard-U corrected DFT method (see Computational Details). The optimized lattice parameters, charge, and magnetic ordering for the employed structures agree well with known experimental bulk values within a 5% difference, tabulated in Supplementary Table 1. Our DFT results show that only $Fe_3O_4$ is energetically more favorable in the inverse (I) spinel ordering ($B[AB]O_4$), while all others prefer the normal (N) spinel ordering ($A[B]_2O_4$), where $A^{+2}$ ions occupy tetrahedral interstices of the oxygen lattice and the $B^{+3}$ ions octahedral interstices in case of normal (N) spinel. The obtained lowest total energy for various bulk spinel oxides $E_i^{bulk}\left(M_3^i O_4\right)$ is subsequently used to normalize the mixing enthalpies introduced later in the context of HEOs.

The binary combinations from the considered elements behave similarly, as only Fe-containing spinel oxides show a propensity towards inverse spinel, most notably $V^{+2}Fe^{+3}$ and $Ni^{+2}Fe^{+3}$ (Supplementary Fig. 1). These findings are also in good quantitative agreement with the known experimental values and crystal field theory predictions (Supplementary Fig. 2)[80,81]. Our calculations of a limited set of ternary spinel oxides of $Co_3O_4$(N), $Fe_3O_4$(I), and $Ni_3O_4$(N), indicate that N↔N mixtures are always preferred over I↔N mixtures (Supplementary Fig. 3). However, generalized bulk DFT models beyond ternary spinel oxides are prohibitively difficult to be constructed and were not pursued. Instead, we devised a simplified HEO impurity model to study these systems.

### Calculation of surface models

For pure spinel systems, we cut spinel bulk structures to $2 \times 2$ unit cells of 100-A surfaces studied previously for $Co_3O_4$[50], allowing octahedral sites to be in the surface layer while constraining the bottom half during the optimization. To model HEO systems, we have inserted the HEO impurity (see Results section) into the same $Co_3O_4$ (100)-A surface. To calculate the OER activity of active sites in HEO systems, we checked the effect of the closest neighboring elements by using 5*5 permutations ($_5P_5$). We used a 15 Å vacuum gap between two slabs for surface calculations to avoid interaction between periodically repeated slabs. A dipole correction was applied to reduce the interactions between the periodically repeated slabs. The surface structures were optimized using a $5 \times 5 \times 1$ Monkhorst–Pack grid with a cutoff energy of 500 eV. The convergence criteria of the electronic self-consistency loop were set to be $10^{-5}$ eV and 0.02 eV/Å for energy and force.

For both pure and HEO spinel active sites, we evaluated the OER activity by calculating the O* and OH* binding energies. The theoretical overpotentials for each active site of pure spinel and HEO were calculated using the standard OER mechanism, which has been applied to many types of oxides (* → OH*, OH* → O*, O* → OOH*, OOH* → $O_2$ (g)). We calculated *OOH adsorption energies for pure spinel and randomly selected 20 HEO surfaces, and those results yielded a scaling relation of OOH* = (0.8318 × OH* + 3.1054), which was used for the rest of the HEO calculations (Supplementary Fig. 21). Using these descriptor values, for each surface, we have calculated O-OH*, and OH* energies and place them on the OER 2D map which is constructed from the function of Max (OH, O-OH, OH + 1.23). Each active site has 120 unique combinations and overpotentials. The colors bar in the heatmap indicates the region on each overpotential. The Gibbs free energies of each step of OER were calculated via the computational hydrogen electrode method with zero-point energy correction. The vibrational enthalpy and entropy contributions were obtained by the quantum mechanical harmonic approximation at room temperature. For the adsorbate reference energies, we used $H_2O$ (l) and $H_2$ (g) only as the energy of $O_2$ in the gas phase is poorly described by DFT calculations.

### Synthesis of spinel oxides and HEOs

All chemicals are used as purchased. Chromium(III) chloride hexahydrate (96%, Sigma Aldrich), Manganese(II) chloride tetrahydrate (ReagentPlus®, ≥99%, Sigma Aldrich), Iron(II) chloride hexahydrate (Reagent grade, 98%, Sigma Aldrich), Nickel(II) chloride hexahydrate (ReagentPlus®, Sigma Aldrich), and Cobalt(II) chloride (97%, Sigma Aldrich) were used as precursors. Absolute ethanol (99.5 + %, Acros Organics) was used as a solvent. Potassium hydroxide (Reagent grade, 90%, Sigma Aldrich) was used as an electrolyte.

We synthesized all the spinel oxides studied here using the reported facile and fast sol-gel flame method[48,49]. First, all precursors for metal cations with equimolar concentration (0.1 M) were dissolved in ethanol and mixed through sonication for 30 min. Next, the precursor solution was spin-coated on clean fluorine-doped tin oxide glass (FTO/glass, 7–8 ohm/sq, MSE supplies) substrates at 3000 rpm for 20 s and then annealed at 150 °C on a hot plate. This process was repeated twice to achieve good coverage and desired thickness (~50 nm depending on the roughness of the FTO/glass substrate). Finally, the precursor-coated FTO/glass substrate was annealed in a post-flame region. The flame annealing was conducted using a co-flow premixed flat flame burner (McKenna Burner), operating with premixed $CH_4$ (fuel) and air (oxidizer) with a fuel/oxygen equivalence ratio of 1.1. The substrate was annealed at 3 cm above the flame for 30 s with a local gas temperature of about 1000 °C, measured by a K-type thermocouple (1/16 in. bead size, Omega Engineering, Inc.). The substrate was subsequently removed from the flame to cool down to room temperature naturally. This rapid high-temperature heating and cooling process could facilitate homogeneous multi-elemental mixing.

### Materials characterizations

The crystal structure of the synthesized samples was investigated by X-ray diffraction (XRD, PANalytical Empyrean) with a Cu source (Kα, 1.54 Å of the wavelength). In-situ temperature-dependent XRD (PANalytical Empyrean HTK 1200 N) was used to explore the crystallinity development in the temperature range of 100–1000 C° under the ambient condition. The sample morphology was characterized using scanning electron microscopy (SEM, FEI Magellan 400). The surface analysis with an elemental quantification was examined using X-ray photoelectron spectroscopy (XPS, PHI VersaProbe 3) that uses a monochromatized Al Kα radiation (1486 eV). The high-resolution images, selected area electron diffraction (SAED), dark-field image, and energy-dispersive X-ray spectroscopy (EDS) elemental mapping images were collected using a scanning transmission electron microscope (STEM, JEOL JEM ARM 200 F). Electron energy loss spectroscopy

(EELS) was obtained by STEM (FEI Tecnai G2 F20 X-TWIN) with a Gatan Tridiem spectrometer with an energy resolution of 0.8 eV. The convergence and collection semi-angle of the EELS detector were 15.5 and 21.8 mrad. To investigate the bulk valence states of Fe and Co, X-ray absorption spectroscopy (XAS) was conducted in transmission mode on electrodes sealed under a pouch under argon at beamline 2-2 at SSRL. A Si (220) φ = 90° monochromator was used and was detuned to 50–60% of the maximum intensity to eliminate higher-order harmonics. The spectra of Fe reference foils were used to calibrate the photon energy by setting the first crossing of zero of the second derivative of the absorbance spectrum to be 7112 eV. Three ion chambers were used in series to measure $I_0$, $I_{sample}$, and $I_{ref}$. Spectrum normalization and alignment were performed using the Athena software package3. XAS analysis and simulation were performed using the Artemis software package. For ICP-MS, samples and standards were gravimetrically prepared to facilitate accurate dilution correction calculations. KOH solutions were dried down and then dissolved in 2% high-purity nitric acid. Analysis was performed on an Agilent 8900 ICP-MS attached to an Agilent SPS-4 autosampler. The sample introduction system also includes a standard Scott double pass cooled spray chamber operated at 2 °C, a 2.5 mm inner diameter of Agilent glass torch, and a 400 μl/ min Micromist nebulizer. The ICP-MS was fitted with standard nickel cones and operated in $NH_3$ mode (4 mL/min) to reduce the presence of isobaric interferences from argides generated in the plasma. Samples were further diluted by 50% with an Sc internal standard teed into the sample introduction system; Sc was used to correct for instrumental drift for the analysis.

## Electrochemical measurement

All the electrochemical measurements, including linear sweep voltammetry (LSV), cyclic voltammetry (CV), chronopotentiometry (CP), chronoamperometry (CA), and electrochemical impedance spectroscopy (EIS), were conducted using a potentiostat (Model Interface 1010, Gamry) in a standard three-electrode electrochemical cell using 1 M KOH (pH ~14) as the electrolyte and the pH value of electrolyte was determined by using Mettler Toledo F20-Kit Benchtop pH Meters. Before measuring the pH of the electrolyte, the pH meter was calibrated with three calibration buffer solutions (pH 4, 7, and 10). The synthesized spinel oxides (from binary to senary oxide) served as a working electrode with a Pt counter electrode and an Ag/AgCl reference electrode (Sat. KCl). The measured potentials versus Ag/AgCl are converted to the potentials versus reversible hydrogen electrode (RHE) by the equation below:

$$E_{RHE} = E_{Ag/AgCl} + 0.059 \times pH + 0.197 \quad (4)$$

All LSV curves were measured at a scan rate of 10 mV s$^{-1}$, and then the potential was corrected by 90% IR compensation to remove the solution resistance effect. The overpotential (η) was determined by η = $E_{RHE}$ − 1.23 (V). The LSV curves were further used to construct the Tafel plots, which plot overpotential (η) as a function of the current density log (j). The EIS raw data were simulated in terms of the equivalent circuit model, depicted in Supplementary Fig. 13. This measurement was conducted between 20 kHz and 0.05 kHz at 1.54 V vs. RHE for all samples. The stabilities of the samples were evaluated both by CP for 7 days (168 h) at the initial current density of ~10 mA cm$^{-2}$ and by CV for 1000 cycles in the potential window ranging from 0.9 to 1.6 V vs. RHE with a scan rate of 100 mV s$^{-1}$. Electrochemical surface area (ECSA) was measured using CV in non-Faradaic regions with different scan rates, and calculated by the equation, ECSA = $C_{dl}/C_s$, where $C_s$ is the specific capacitance, which is usually used to be about 20 – 60 μF·cm$^{-2}$ in alkaline solutions. We took 40 μF·cm$^{-2}$ as the $C_s$ value. The evolved oxygen was measured by the oxygen fluorescence sensor (Neofox, Ocean optics) and used to calculate the Faradaic efficiency by comparing with the calculated amount from the CA curve

at 1.5 V vs. RHE for 2 h as follows:

$$FE = \frac{n \times N_{O_2} \times F}{I \times t} \quad (5)$$

where n, $N_{O_2}$, F, I, and t represent the number of electrons (i.e., 4), the mole of oxygen measured, the Faradaic constant, measured current, and time, respectively.

## Data availability

The complete list of calculated structures, their total and adsorption energies, and DFT settings can be found at https://www.catalysis-hub. org/publications/HossainInvestigation2022 on Catalysis-hub.org platform[82].

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

## Acknowledgements

This research was supported by the U.S. Department of Energy, Office of Science, Office of Basic Energy Sciences, Chemical Sciences, Geosciences, and Biosciences Division, Catalysis Science Program to the SUNCAT Center for Interface Science and Catalysis. Use of the Stanford Synchrotron Radiation Lightsource, SLAC National Accelerator Laboratory, is supported by the U.S. Department of Energy, Office of Science, Office of Basic Energy Sciences under Contract No. DE-AC02-76SF00515. X.L.Z. would like to thank the National Science Foundation EFRI-DCheM program (Agreement Number: SUB0000425) for their generous support. J. B. would like to acknowledge the Chevron Fellowship in Energy. Part of this work was performed in part in the Stanford SIGMA Facility with support from the Stanford Doerr School of Sustainability and the Stanford Nano Shared Facilities (SNSF)/Stanford Nanofabrication Facility (SNF), supported by the National Science Foundation under award ECCS-2026822. The authors would like to acknowledge the use of the m2997 computer time allocation at the National Energy Research Scientific Computing Center, a DOE Office of Science User Facility supported by the Office of Science of the U.S. Department of Energy under Contract No. DE-AC02-05CH11231.

## Author contributions

J. B., M. D. H., M. B., and X. L. Z. conceived the ideas and designed the project. M. B. and X. L. Z. supervised this project. J. B. carried out the experiments including material synthesis, characterizations, and electrochemical measurements. M. D. H. conducted computational investigations. P. M. contributed to the material characterizations (TEM, EELS, and EDS). J. L. and W. C. C. contributed to the X-ray absorption spectroscopy measurement and analysis. K. T. W. participated in the project discussion. J. L. and Y. J. contributed to the material characterizations. J. B. and M. D. H. wrote the draft. J. B., M. D. H., M. B., and X. L. Z. edited the manuscript and finalized it. All authors participated in the discussion and editing.

## Competing interests

The authors declare no competing interests.
