## [Peer Review File · Nature Communications]

REVIEWER COMMENTS

Reviewer #1 (Remarks to the Author):

This manuscript reports a high-entropy spinel oxide as OER catalyst. Much effort has been put into the theoretical study of the thermodynamic stability and simplified HEO impurity model construction. The synthesis, characterization and electrochemical tests are also well illustrated subsequently. However, the manuscript still needs more solid experimental evidence and deeper insights to improve its quality. Therefore, I suggest a major thorough revision before considering its acceptance, and some questions are listed below.

1. It can be noticed in table S4 that a $(\text{CoCrFeMnNi})_3\text{O}_4$ material reported in ref. 15 has a chemical formula identical to this manuscript and exhibits an even lower overpotential than this article. What is the difference and advantage of this report?

2. The real occupation of different cations in the spinel structure is critical to examine the reliability of the DFT modelling in this article. Although it is mentioned that A^{2+} ions occupy tetrahedral interstices and the B^{3+} ions octahedral interstices in normal spinels, the situations for different metals are different in reality. For example, the calculated octahedral site preference energy shows that Ni^{2+} prefers the octahedral site (see for example J. Am. Chem. Soc. 2020, 141, 8136). Hence, it is necessary for the authors to substantiate the occupation of different metals in the random HEO system.

3. What is the origin of the XPS peaks of Sn 2p emerged in figure 8c and 9a? Particularly, the positions of the Sn 2p peaks of sample Fe and CoFeNiMn are rather different with the others, what's the reason? The authors should fit the XPS profiles more scrupulously as it seems that the Sn 2p peak coincides the satellite of Fe 2p.

4. It is stated that "HEO has higher electrical conductivity than the lower entropy oxides" (page 9) as suggested by the Nyquist plots fitted with a REAP2CPE equivalent circuit model (figure S12) which is different with most literatures. The authors should cite the references to illustrate the reliability of this equivalent circuit model.

5. What is the meaning of the word "charge" in sentence "It should be noted that part of the charge is related to surface coverage by generated oxygen bubbles..."?(Page 9) Is it a spelling mistake of "change"?

6. Figure 3e shows that the oxidation state of Mn decreases during OER, which is unnormal under oxidative potentials. Can the authors explain the reason? Dose it indicate the rearrangement of the cation occupation in spinel lattice?

7. As known, the Cr in spinel is prone to dissolve under anodic bias in alkaline medium (see for example, Angew. Chemie Int. Ed. 2021, 60, 7418–7425). The authors should investigate and discuss this behavior of the HEO.

8. This manuscript emphasizes the efficacy of local strain on OER activity based on the DFT calculation of intermediate binding energies, which however lacks experimental evidence for the lattice strain.

9. The page numbers of the reference items are missed.

Reviewer #2 (Remarks to the Author):

The paper by Baek et al describes the combined theoretical and experimental study of spinel high-entropy oxides (HEO's) for the oxygen evolution reaction. A theoretical impurity model is used to evaluate the activity and stability of HEO active sites, and the material is synthesized and characterized experimentally. The HEO is found to have superior properties compared with oxides containing fewer elements. This is indeed an interesting conclusion, however to fully support it and make the paper suitable for publication in Nature Communications some details must be clarified as outlined below:

Major comments:

1) The model considers the 5 nearest atoms, however from the picture presented in fig 1a it is difficult to see that these are indeed the 5 nearest atoms. A top view should be added to help the reader.

2) It is stated that there are 120 distinct permutations. How does the number 120 come about? Does this mean that the location of the different elements relative to the active site is assumed to be unimportant? If so this should be clearly stated. Has the error associated with this assumption been quantified?

3) Some points in Figure 4a are quite far away from other points of the same elements – indeed the most active Co and Fe sites appear to be outliers. Can it be explained why these sites are so good? How were outliers checked and possibly removed or constrained to avoid errors in the calculations?

4) From the experiment it is concluded that some FeNi-OOH is formed. The calculations show that the overpotential of FeNi-OOH is higher than for the best HEO sites, however only 5% of the HEO sites have overpotentials below 0.4V, while the surface of FeNi-OOH have identical sites with an overpotential of 0.37V. Can you be sure that the observed activity stems from the HEO and not at least partly from FeNi-OOH?

Minor comments:

- Abstract, L27-28: Incorrect sentence

- Methods section: Some information is missing: Which values of U were employed and how was this chosen? How many layers are in the surface models?
- Methods section: What is meant by “5*5 permutation cells”?

Reviewer #3 (Remarks to the Author):

In this paper, the authors reported the high entropy oxide (HEO) based on Co, Fe, Ni, Cr, and Mn prepared using a rapid sol-flame method. The authors use theoretical and experimental approaches to scrutinize the OER activity and stability for the obtained spinel-type HEOs. The manuscript can be reconsidered for publication after addressing the following comments.

1. the authors claim “The X-ray diffraction (XRD) of the resulting HEO in Fig. 2b shows a pure spinel Fd-3m (227) crystal structure with ICDD 00-01-0325.” However, there is an obvious diffraction peak at around 33°, which does not belong to the Fd-3m (227) spinel crystal structure. Likewise, such a impurity peaks can be seen in Fig. S5. This has undermined the discussion
2. In page 6 and 7, please double check the d-spacings for their corresponding diffraction planes carefully for both the synthesized sample and the reference NiFe₂O₄.
3. The reduce peak intensities in the Cr 2p and Fe 2p XPS spectra with the catalysis time indicates the leaching of the metals during electrolysis. The authors are advised to provide data to show the metal leaching during OER, otherwise, please explain why the peak intensity is reduced.
4. Please spell “EC” that appears in “EC measurement” in page 10.
5. The authors say that “... and divalent state in Mn increase within a couple of hours of the test.” However, visual inspection of the ex-situ XPS Mn 2p spectra (Fig. S16) suggests positive slight of the spectra with the time. Please double check and clear this paradox.
6. “Detailed XPS analysis in Fig. S16 indicates that the Fe³⁺ and Ni³⁺ and OH concentrations were increased at the surface for 5 hours. This result demonstrates that the surface Fe-NiOOH was formed during the EC measurement, which influenced the slight change of overpotential (Fig. 3e). Although it is well known that the surface oxyhydroxide is also a promising OER catalyst, HEO still plays an important role as support contributing to the great stability.” The question is that why only Fe-NiOOH formed during the electrochemical measurement? What happened to the other metals? In other words, solid evidence and discussion must be provided to support the statement above.
7. Why does the existence of Mn lead to degraded performance?

Reviewer #4 (Remarks to the Author):

Manuscript NatComm_379092_0

General comments:

This work deals with high entropy oxides (HEO) as OER catalysts. The work is an attempt to link DFT calculation with experiments. One of the main claims is that HEOs are better OER catalysts than mono, binary etc. catalysts. The reason would be that Co^{3+} is the active site, based on DFT and material characterizations.

HEOs are a very complex class of materials for which establishing a clear cut structure and composition – OER performances is very challenging. The work is of quality but is nevertheless not compelling enough and it presents severe limitations as described below.

About the approach:

- (i) the structural and microscopic characterizations are only conducted on the as-prepared HEO and other oxides. This is very annoying. Authors indeed conclude that there is more Co^{2+} and more Fe^{3+} close to the surface while they claim Co^{3+} is responsible for the OER activity. Authors must be aware that the surface region of most OER catalysts is known to undergo structural changes (see more below). Therefore, operando characterizations (for instance XRD, XAS are quite feasible) or, at least, presenting detailed post-operation characterizations appears indispensable.

- (ii) the analysis of electrochemical data is superficial. Authors never address the key question of the normalization of the OER current while this is a prerequisite to compare different catalysts. Many works have addressed this difficult question and these works are not cited. Proper benchmarking of samples needs to be conducted.

- (iii) the coupling between DFT and experiments appears very weak (see more details below). In addition, as stated above, the real OER active surface is probably not a (001) facet.

About the claims:

- HEO are claimed to be better performing than mono, binary etc. oxides. This conclusion is biased. Reality is much more complex than this. In fact, the performances of binary oxides depend on their composition and at optimum composition it may be better than that of corresponding mono

element oxides. See literature about CoFe and NiFe oxides: In both cases, an optimum composition that departs from 50/50 enables reaching a better OER activity than the corresponding mono-oxides.

- DFT suggests that Co³⁺ sites are activated at low overpotential while a larger overpotential is required to activate Cr and Fe sites. In the first place, it is not convincing that the experimental situation (current < 10 mA/cm²) fits the conditions of low overpotential in DFT. Second, Co sites are in sufficient proportion at the surface to be characterized in operando. Last, XPS and EELS analysis suggest the present of Co²⁺ at the surface.

- Role of local strains at Co sites. Again, authors should provide experimental data in support of this statement. As said before, the structural characterizations are not sufficiently coupled with DFT. If strains are present at the surface of the grains, these should be detectable by operando XRD, because, after all, Co sites must occupy a good fraction of all surface sites.

Recommendation:

In its present form the work is not recommended for publication and it should be rejected. The experimental part is rather a routine applied material science study in relation with OER. Authors should provide compelling experimental data that support that HEOs present better intrinsic OER activity. Relying on DFT is very dangerous since the calculation is performed on a perfect (001) plane while the real surface of the HEO likely undergoes a restructuring as many OER catalyst do. In addition, recent experimental and ab initio studies suggest that the OER active site may be composed of 2 atomic sites.

Specific comments:

1) DFT

- Experimentally, the synthesis of HEOs relies on rapid thermal quenching of a melt. Should not it be expected that the structure is metastable? On which basis is calculated the data in Fig. S2 which gives the crystal field energy of materials. Is the calculation considering a structure at equilibrium?

- It is extremely difficult to understand how the underlying 2D volcano plot in Fig. 4a has been constructed.

- The graph in Fig. 4b is confusing. Authors must better explain why only a fraction of sites are activated at a given overpotential. This must have some relation with Figs. S17 and S18. In addition, it is recommended plotting, for each element the % of activated sites (normalized to the TOTAL

number of surface sites of this element). A vertical log scale would be useful. With one graph per element one could catch which type of surface site is most active at a given overpotential.

- The discussion of the role of local strain on activity of Co sites is not very convincing in the present form (bottom of page 13). Authors should provide experimental data in support of this statement. If strains are present at the surface of the grains, these should be detectable by operando XRD or XANES, because, after all, Co sites represent a good fraction of all surface sites.

- Fig. S4: It is very difficult to catch the message from this figure. A complete description (in SI) is mandatory to explain this figure.

2) Material characterizations:

Authors performed characterizations using different techniques. However, the data analysis remains rather limited in each case. Structural / morphological data of mono, binary ... HE oxides are missing to judge whether the comparison of catalysts is sound.

- The average thickness of layers is not mentioned. What is the “desired thickness” in page 18?

- XRD: page 5, line 161 and Fig. S5: the term “in situ” refers to temperature dependent XRD measurements. There might be a confusing with in situ (i.e. in electrochemical environment) XRD. Authors should determine the grain size from their data and compare it with TEM images. Showing similar data for mono, binary etc. oxides is mandatory to compare the ECSA (see comments below).

- XANES: Page 7, lines 190-195 and Fig. S10: the analysis is too superficial. Considering the slope of the XANES plot is definitely not appropriate. The edge energy is the relevant parameter and it must be quantitatively measured and converted into an average oxidation state using the calibration plot. The calibration plot must be presented and authors must give values (with error bar) of the Edge energies and corresponding average oxidation state of elements. Authors perfectly know that XAS gives the average oxidation state of elements. Therefore, additional characterizations and analysis are required to gain insights into the valence state of cations at the surface.

- EELS: Fig. 2g-k and related text, page 7: Was EELS at different locations performed on other particles (how many?) to make sure that this is representative. In (h) the signal is rather noisy and one wonders about the precision of the integration of lines L2 and L3 (especially for Fe). All this should be better documented. Error bars are also requested. In Fig. 2i-j one would like to see plots of

the intensity of individual lines L3 and L2 of all elements (with error bars) and not just the ratio L3/L2 for Fe and Co. Authors conclude that there is more Co²⁺ and more Fe³⁺ close to the surface while they claim Co³⁺ is responsible for the OER activity. Explain the discrepancy. This points to the need of operando spectroscopic characterizations.

- XPS:

Page 7, line 171: XPS probes the near surface composition. You cannot state it gives the surface composition. For instance, it is impossible from the data to state that Co³⁺ or Co²⁺ is that the surface and would act as active site. A reader would like to have the information in Table S2 closer to the main text.

Page 7, lines 183 - 185: "This implies that HEO has more Fe³⁺ and Co²⁺ at the surface, elaborated by electron energy loss spectroscopy (EELS) later". The reasoning leading to this statement is obscure and the formulation looks strange. Once again, the claim in abstract is that Co³⁺ are the active site. Explain the discrepancy.

Figs. S8 and S9: Obviously, the decomposition of spectra is not unique, which implies the determination of the relative concentrations of valence state per element cannot be given with the precision given in Table S2. Provide error bars. In addition, authors should give the atomic ratio of the different elements to judge whether there is a difference between the surface and the bulk composition.

3) Electrochemical analysis:

It is very annoying that the authors do not make any attempt to normalize the current to a sensible parameter such as the electrochemical active surface area (ECSA). A normalization of the OER current by the geometrical surface area of the substrate may be fine for application oriented approach but it tells nothing about the intrinsic OER activity because the samples present a complex morphology and are perhaps porous. Not only, the electrochemically active surface area (ECSA) may be quite larger than the geometrical area but it may also vary from sample to sample.

The set of data does not give enough information pertaining that the ECSA may be similar for all mono, binary ... HE oxides used in this study. A change in ECSA by a factor 3 is, for instance, enough to modify the overpotential by 15 mV to 30 mV, according to the Tafel slope.

Table S4. This Table cannot be used to state that HEO are intrinsically better OER catalysts. A true benchmarking analysis is missing.

The Tafel slope of CoO appears larger than previously reported for Co₃O₄.

Fig. 3c: this graph is rather misleading since a mono cation oxide (e.g. CoO) may be performing better than those reported here. In addition, binary oxides may perform better than mono oxides after optimization of the composition. The composition of different oxides should be given in the figure.

The equivalent electrical scheme used to fit EIS must be justified. In addition, fitting procedure must be described because with so many parameters it is hard to believe that one can unambiguously determine R_2 , the bulk resistance, by adjusting one spectrum. Why authors do not exploit the other parameters derived from the EIS modelling?

4) Minor comments:

Page 1, line 29: "Such catalyst exhibits the highest OER activity of 309 mV at 10 mA cm⁻²": this is a laboratory language, certainly not a scientific one. Please reformulate.

Point-by-point response letter

New Contribution: We sincerely appreciate all reviewers providing valuable comments. We have conducted additional experimental characterizations and DFT calculations in response to reviewer comments. We believe that addressing the reviewers' comments improved our manuscript significantly. Here is a summary of additional contributions:

- Experimental part
 - Higher resolution of TEM-EELS: Fig. 2g-2j and Supplementary Fig. 11
 - Microstrain and crystallite size calculations: Supplementary Fig. 27 and Supplementary Table 6
 - ICP-MS: Supplementary Fig. 19
 - Remeasurement of XPS: Supplementary Fig. 7 and 8
 - Average atomic concentrations with error bars from more spots in XPS: Supplementary Table 2
 - ECSA measurement: Supplementary Fig. 14
 - SEM cross-section for thickness measurements: Supplementary Fig. 5
 - Calculation of oxidation states from XANES: Supplementary Fig. 10
 - Detailed EIS description: Supplementary Fig. 13
- Theory Part: HEO comparison, Bader Charge calculation (Fig. R1), Figure 1 & 4 modification.

Our response to reviewer comments will be in **blue color** and the revised parts in **red color**, which are highlighted in **yellow color** in the revised manuscript.

Reviewer #1 (Remarks to the Author):

This manuscript reports a high-entropy spinel oxide as OER catalyst. Much effort has been put into the theoretical study of the thermodynamic stability and simplified HEO impurity model construction. The synthesis, characterization and electrochemical tests are also well illustrated subsequently. However, the manuscript still needs more solid experimental evidence and deeper insights to improve its quality. Therefore, I suggest a major thorough revision before considering its acceptance, and some questions are listed below.

Response: We appreciate the reviewer's recognition of our effort to study the HEO system theoretically and experimentally. Below, we tried to answer all the raised comments.

1. It can be noticed in table S4 that a $(\text{CoCrFeMnNi})_3\text{O}_4$ material reported in ref. 15 has a chemical formula identical to this manuscript and exhibits an even lower overpotential than this article. What is the difference and advantage of this report?

Response: Thanks for your comment. Though the material reported in **Ref. 15** has the same chemical composition, its morphology and particle size are different from ours. Their average particle size is about 6 nm (**see Fig. 1 in Ref. 15 attached below**), in comparison to 21 nm in our case (**Fig. 2**). The different particle sizes and surface areas cause the difference in the catalytic performance. Along with the particle size, the ratio of oxidation state for different elements also

differs from our result. In our project, our goal is not to optimize the catalytic performance of a particular compound, but to compare the catalytic activity for various entropy oxides from binary to senary. Additionally, we used the solution flame method which enables us to synthesize various oxides rapidly and simplify the comparison of catalytic properties of those different oxides.

[Ref. 15] **Fig. 1** (a) XRD patterns of M3O4-T ($T = 0, 300, 400,$ and 500 C) and the microstructural characterization of M3O4-400: (b) TEM image; (c) particle size distribution histogram; (d and e) HRTEM images

Revision: We have modified Fig. 2 as follows.

Fig. 2 | Characterization methods for as-synthesized spinel HEO. **a** Schematic of the sol-flame synthesis process. **b** XRD and **c, d** high-resolution TEM images of HEO (inset: FFT image). **e** SAED pattern and **f** EDS-mapping images of Co, Fe, Ni, Cr, Mn, and O. **g** Dark-field image of HEO particles and **h** EEL spectra extracted from the four points with a distance of 1.5 nm between each point (0.5~5 nm) in **g**, showing O-K edge and L2,3-edges of Co, Fe, Ni, Cr, and Mn. **i** Fe L3/L2 and Co L3/L2 white-line intensity ratio for the four points (0.5~5 nm) in **g**. **j** The energy difference between O K-edge pre-peak and main peak (energy separation, ΔE) at the four points (0.5~5 nm) in **g**.

2. The real occupation of different cations in the spinel structure is critical to examine the reliability of the DFT modelling in this article. Although it is mentioned that A^{2+} ions occupy tetrahedral interstices and the B^{3+} ions octahedral interstices in normal spinels, the situations for different metals are different in reality. For example, the calculated octahedral site preference energy shows that Ni^{2+} prefers the octahedral site (see for example J. Am. Chem. Soc. 2020, 141, 8136). Hence, it is necessary for the authors to substantiate the occupation of different metals in the random HEO system.

Response: The main finding in the reference article [J. Am. Chem. Soc. 2020, 141, 8136] is that Fe dopant preferably entered the tetrahedral site of their studied $NiCo_2O_4$ spinel system. This is consistent with our studies about the stability of binary (Supplementary Figs. 1 and 2) and ternary (Supplementary Fig. 3) spinel systems. For spinel oxides with more than 3 metals such as HEO, we have statistically tested this observation with random occupations, and our stability distribution shows that the most stable occupations have Fe^{+2} in tetragonal sites.

In our impurity model, each active site (Co, Fe, Cr) is bonded with 5 neighbors (3 octahedral sites and 2 tetrahedral sites). When we check the activity for each active site, we place e.g., Ni into 2 tetrahedral and 3 octahedral sites. This process is repeated for each element. In the end, for each active site, we ended up with 120 distinct calculated clean surfaces, 120 O_{ads} , 120 OH_{ads} energies, selected OOH^* energies, and plotted the overpotential in the heat map. We found that Co^{3+} , Cr^{3+} , and Mn^{3+} are preferred octahedral sites while Fe^{2+} and Ni^{2+} are comparatively preferred to occupy the tetrahedral sites for the most stable configurations.

3. What is the origin of the XPS peaks of Sn 2p emerged in figure 8c and 9a? Particularly, the positions of the Sn 2p peaks of sample Fe and CoFeNiMn are rather different with the others, what's the reason? The authors should fit the XPS profiles more scrupulously as it seems that the Sn 2p peak coincides the satellite of Fe 2p.

Response: We agree with the reviewer's concern. As shown in the previous data (Supplementary Figs. 7 and 8), the large peak of Sn 2p makes it difficult to deconvolute the Fe 2p_{3/2} region, especially the Fe 2p_{3/2} satellite peak. This is because we measured HEO on FTO/glass substrate which causes the Sn 2p peak to overlap with Fe 2p region. As the reviewer said, we need a more accurate Fe 2p peak for a more scrupulous analysis. We have switched all the XPS results even in other elemental regions and there is no more Sn 2p peak interfering with the interpretation of the Fe 2p region from which we could get more accurate deconvoluted peaks and concentrations of elements/oxidation states. Finally, we found that the concentration of Fe^{2+} is more than Fe^{3+} at the

surface of the oxide. We have switched the previous plots to all plots without interference from the peak from the FTO substrate and mentioned the accurate information in the manuscript.

(Previous data) Supplementary Fig. 7 | Detailed XPS elemental survey of HEO particles. a, Cr 2p, **b,** Mn 2p, **c,** Fe 2p, **d,** Co 2p, **e,** Ni 2p, and **f,** O 1s. Cr has only a single oxidation state (Cr³⁺) with peaks between 575.7 and 578.9 eV in the Cr 2p_{3/2} region. The Mn 2p_{3/2} region consists of the peak at 640.1 eV corresponding to the divalent state, while the peak at 642.2 eV belongs to the trivalent state. In the Fe 2p_{3/2} region, the peak at 709.7 eV indicates a divalent state, while the peak at 711.2 eV shows the existence of the trivalent state. The Co 2p_{3/2} spectra exhibit the Co³⁺ peak at 779.6 eV and Co²⁺ peak at 781.6 eV. Lastly, the Ni 2p_{3/2} has the dominant Ni²⁺ peak at 854.9 eV and the inferior Ni³⁺ peak at 856.5 eV. The valence state positions of all elements show a little deviation from the literature values as the HEO particle incorporates random bonding distances between the atoms.

(New data) Supplementary Fig. 7 | Detailed XPS elemental survey of HEO particles. **a**, Cr 2p, **b**, Mn 2p, **c**, Fe 2p, **d**, Co 2p, **e**, Ni 2p, and **f**, O 1s. Cr has only a single oxidation state (Cr³⁺) with peaks between 575.7 and 578.9 eV in the Cr 2p_{3/2} region. The Mn 2p_{3/2} region consists of the peak at 640.1 eV corresponding to the divalent state, while the peak at 642.2 eV belongs to the trivalent state. In the Fe 2p_{3/2} region, the peak at 709.7 eV indicates a divalent state, while the peak at 711.2 eV shows the existence of the trivalent state. The Co 2p_{3/2} spectra exhibit the Co³⁺ peak at 779.6 eV and the Co²⁺ peak at 781.6 eV. Lastly, the Ni 2p_{3/2} has the dominant Ni²⁺ peak at 854.9 eV and the inferior Ni³⁺ peak at 856.5 eV. The valence state positions of all elements show a little deviation from the literature values as the HEO particle incorporates random bonding distances between the atoms.

(Previous data) Supplementary Fig. 8 | XPS elemental survey result of Fe 2p and Co 2p for different spinel oxides. a, Fe 2p, and b, Co 2p to compare the amount of each oxidation state for multi-element spinel oxides.

(New data) Supplementary Fig. 8 | XPS elemental survey result of Fe 2p and Co 2p for different spinel oxides. a, Fe 2p, and b, Co 2p to compare the amount of each oxidation state for multi-element spinel oxides.

Revision: (Page 7)

“They exhibit different $\text{Fe}^{2+}/\text{Fe}^{3+}$ and $\text{Co}^{2+}/\text{Co}^{3+}$ at the surface where HEO shows the higher $\text{Fe}^{2+}/\text{Fe}^{3+}$ and lower $\text{Co}^{2+}/\text{Co}^{3+}$ than the lower entropy oxides. This implies that HEO has more Fe^{2+} and Co^{3+} at the surface, elaborated by electron energy loss spectroscopy (EELS) in the later section. Thus, the different ratios of divalent and trivalent states in Co and Fe 2p_{3/2} influence the OER activity. Also, the average concentrations of trivalent and divalent states of all elements for different spinel oxides are summarized in **Supplementary Table 2.**”

4. It is stated that “HEO has higher electrical conductivity than the lower entropy oxides” (page 9) as suggested by the Nyquist plots fitted with a REAP2CPE equivalent circuit model (figure S12) which is different with most literatures. The authors should cite the references to illustrate the reliability of this equivalent circuit model.

Response: In our work, the impedance raw data were simulated in terms of the circuit model known as the Armstrong-Henderson equivalent circuit (depicted in **Supplementary Fig. 13**). In this equivalent circuit, R1 represents the uncompensated solution resistance, R2 is the charge transfer resistance, and the resistance R3 represents the parameter associated with the relaxation of the surface coverage of an adsorbed intermediate in the OER mechanism. As the reviewer suggested, we added the proper references to the manuscript.

Revision: (Page 19, Electrochemical measurement section)

“The EIS raw data were simulated in terms of the equivalent circuit model known as Armstrong-Henderson equivalent circuit, depicted in **Supplementary Fig. 13.**^{62–64}”

5. What is the meaning of the word “charge” in sentence “It should be noted that part of the charge is related to surface coverage by generated oxygen bubbles...”? (Page 9) Is it a spelling mistake of “change”?

Response: We apologize for the mistake and have changed “charge” to “change”.

6. Figure 3e shows that the oxidation state of Mn decreases during OER, which is unnormal under oxidative potentials. Can the authors explain the reason? Dose it indicates the rearrangement of the cation occupation in spinel lattice?

Response: We thank the reviewer for this comment. Yes, indeed, the decrease in the Mn oxidation state must be due to the rearrangement of the cations within the HEO structure.

We know that spinel oxides are composed of tetrahedral and octahedral sites. The bulk Mn_3O_4 structure has the most stable normal spinel structure which consists of 2 tetrahedral sites (occupied by Mn^{2+}) and 4 octahedral sites (occupied by Mn^{3+}) (**Fig. R1a**). The shown Bader charge analysis and outer shell electron numbers in **Fig. R1b** confirm these occupations. Next, our HEO impurity model results show that the most stable HEO configurations have Fe^{2+} and Ni^{2+} in the tetrahedral sites, and Co^{3+} , Mn^{3+} , and Cr^{3+} are in the octahedral sites (**Fig. R1c**). However, we have also observed that Fe^{2+} and Ni^{2+} are gradually leached and out from tetrahedral sites during electrochemical conditions likely forming oxidized oxyhydroxides (**Fig. 3e**). At the same time, we have observed Mn^{3+} to be reduced to Mn^{2+} and possibly moving to the now empty tetrahedral sites (**Fig. R1d**). This rearrangement of Mn makes the HEO system $\sim +0.5$ eV less stable than at the former position (octahedral site). As these rearrangements are happening under applied potential conditions thus quite highly possible. On the other hand, the reduction of Cr^{3+} to Cr^{2+} to tetrahedral sites is relatively uphill ($\sim +0.95$ eV) than Mn^{3+} to Mn^{2+} .

Fig. R1. Elemental re-arrangement within the HEO system. **a** Bulk structure of Mn_3O_4 spinel system. **b** Corresponding Bader charge and outer electrons for bulk Mn_3O_4 . **c** Most stable HEO system as compared to the system with Mn^{2+} in the tetrahedral site (**d**).

7. As known, the Cr in spinel is prone to dissolve under anodic bias in alkaline medium (see for example, *Angew. Chemie Int. Ed.* 2021, 60, 7418–7425). The authors should investigate and discuss this behavior of the HEO.

Response: We agree that Cr dissolves over time as shown in the XPS results in **Supplementary Fig. 18**. As the reviewer suggested, we investigated how much the constituent elements were dissolved to the electrolyte over time using inductively coupled plasma mass spectrometry (ICP-MS) in **Supplementary Fig. 19**. We found that ~ 400 ng/ml of Cr and ~ 50 ng/ml of Ni dissolved after the electrochemical measurement. Based on the certificate of analysis for the trace elements in water from NIST (**ref. 67**), quite a large amount of Fe is basically detected from pure water, so we believe that Fe did not dissolve after the electrochemical measurement. For the structural changes, we assume that the (oxy)hydroxide species are produced since the OH intensity increased a lot based on the XPS result and Co, Fe, Ni, and Cr can be candidates as their trivalent state increases or exists.

Revision: (Page 11)

“In addition, we investigated how much the constituent elements were dissolved to the electrolyte over time using inductively coupled plasma mass spectrometry (ICP-MS) in **Supplementary Fig. 19**. We found that mostly certain amount of Cr was definitely dissolved which is well-known from the previous report,⁶⁶ and a little amount of Ni was also dissolved. Based on the certificate of analysis for the trace elements in water from the National Institute of Standards & Technology (NIST),⁶⁷ quite a large amount of Fe is basically detected from pure water.”

Supplementary Fig. 19 | Inductively coupled plasma mass spectrometry (ICP-MS) analysis. ICP-MS result for trace elements in the water reported by the National Institute of Standards &

Technology (NIST) and the dissolved elements in the 1 M KOH electrolyte after the electrochemical measurements (10 min, 1 hour, and 5 hours).

8. This manuscript emphasizes the efficacy of local strain on OER activity based on the DFT calculation of intermediate binding energies, which however lacks experimental evidence for the lattice strain.

Response: Fundamentally, the lattice structure of the multi-element solid solution phase (of HEO) is distorted, which leads to microstrain. To provide experimental evidence, we measured the XRD for some of the spinel oxides including HEO and used these XRD patterns to obtain the Williamson-Hall plot, which is used to calculate the crystallite size and microstrain of each spinel oxide (**Supplementary Fig. 27 and Supplementary Table 6**). Here, we used the Scherrer equation to estimate the grain size. Here, we used the Scherrer equation and Williamson-Hall plot to estimate the grain size and microstrain,

$$D = \frac{K\lambda}{\beta \sin \theta}$$

, where D is grain size, K = 0.9 (Scherrer constant), $\lambda = 0.15406$ nm (wavelength of the x-ray sources (Cu $K\alpha$)), β = FWHM (in radians), and θ = peak position (in radians).

In the Williamson-Hall plot,

$$\beta \sin \theta = 4\epsilon \sin \theta + \frac{K\lambda}{D}$$

, where ϵ = microstrain.

As shown in **Supplementary Fig. 27**, HEO shows the highest microstrain among different spinel oxides. We added this in the Supplementary Information and described it in the manuscript.

Revision: (Page 15)

“In order to provide experimental evidence of the microstrain, we additionally measured XRD for some of the spinel oxides including HEO and employed the XRD patterns to obtain the Williamson-Hall plot, which can be used to calculate the microstrain and crystallite size of each spinel oxide. As shown in **Supplementary Fig. 27 and Supplementary Table 6**, HEO shows the highest microstrain among the various spinel oxides. We assume that, fundamentally, the lattice structure of the multi-element solid solution phase (of HEO) is distorted due to a variety of elemental bonds, which leads to microstrain and is in good agreement with the calculation of the bonding and prediction of HEO activity.”

Supplementary Fig. 27 | Estimation of microstrain and crystallite size from XRD. **a** XRD result for several spinel oxides and **b** calculated crystallite size and microstrain for each oxide. Here, we used the Scherrer equation and Williamson-Hall plot to estimate the grain size and microstrain,

$$D = \frac{K\lambda}{\beta \sin \theta}$$

, where D is grain size, K = 0.9 (Scherrer constant), $\lambda = 0.15406$ nm (the wavelength of the x-ray sources (Cu K α), β = FWHM (in radians), and θ = peak position (in radians).

In the Williamson-Hall plot,

$$\beta \sin \theta = 4\epsilon \sin \theta + \frac{K\lambda}{D}$$

, where ϵ = microstrain.

Supplementary Table 6 | Crystallite size and microstrain for different spinel oxides calculated from XRD results in Supplementary Fig. 27.

	Crystallite Size (nm)	Microstrain
NiFe	42.47	1.03
CoCr	17.26	1.46
CoNiFe	41.49	1.55
CoNiCr	9.64	1.4
CoNiFeCr	12.48	3.03
CoNiFeCrMn	24.02	8.25

9. The page numbers of the reference items are missed.

Response: We added the reference page numbers. Thanks for the correction.

Reviewer #2 (Remarks to the Author):

The paper by Baek et al describes the combined theoretical and experimental study of spinel high-entropy oxides (HEO's) for the oxygen evolution reaction. A theoretical impurity model is used to evaluate the activity and stability of HEO active sites, and the material is synthesized and characterized experimentally. The HEO is found to have superior properties compared with oxides containing fewer elements. This is indeed an interesting conclusion, however, to fully support it and make the paper suitable for publication in Nature Communications some details must be clarified as outlined below:

Response: We appreciate that the reviewer finds our study interesting. Please find the response to reviewer detailed comments below.

1) The model considers the 5 nearest atoms, however from the picture presented in fig 1a it is difficult to see that these are indeed the 5 nearest atoms. A top view should be added to help the reader.

Response: We appreciate for the great suggestion. We have added the top view of our model in the revised Fig. 1.

Fig. 1 | HEO impurity model for high entropy spinel M_3O_4 . **a** The HEO system contains the OER active site shown in the center with adsorbed O^* (red) surrounded by 5 metal neighbors (labeled as A to E for Cr, Co, Mn, Ni, and Fe) within a bond length of 3.5 Å. The inset figure shows the top view of HEO containing active sites (AS) surrounded by neighbor atoms. **b** Relative mixing enthalpy per mixing metals for various active sites of the HEO surface as referred to their bulk systems.

2) It is stated that there are 120 distinct permutations. How does the number 120 come about? Does this mean that the location of the different elements relative to the active site is assumed to be

unimportant? If so, this should be clearly stated. Has the error associated with this assumption been quantified?

Response: For each active site, we assume that the 5 metal cations can be arranged in any order in the 5 neighboring sites (locations). We, therefore, treat each location as different, and ordering in sites is important. For this reason, we performed permutations (not combinations) of 5 elements in 5 sites without repetition, i.e., 5!

$$P(5,5) = \frac{5!}{(5-5)!} = 120$$

The location of different elements is clearly important and shows the different activity of active sites, and we find a wide distribution of binding energy.

Revision: (Page 4)

“The location of five elements is important, which affects active site activity significantly.”

3) Some points in Figure 4a are quite far away from other points of the same elements – indeed the most active Co and Fe sites appear to be outliers. Can it be explained why these sites are so good? How were outliers checked and possibly removed or constrained to avoid errors in the calculations?

Response: We appreciate this important comment. The presence of certain elements causing contraction or expansion leads to improved activity via local strain and slight coordination changes (see below). The outliers are the results of a certain combination of HEO, and we additionally checked by hand for structural and magnetic relaxation convergence. All calculated structures presented here were within the narrow tolerance of structural deformations allowed (< 0.4 Å) from the idealized spinel surface. The calculated data is viewable online at <https://www.catalysis-hub.org/publications/HossainInvestigation2022>.

Further, we explained briefly in our manuscript the origin of these activity outliers in the section ‘**The role of local strain on HEO activities**’.

The active site activity largely depends on the neighboring metals. We found that the bonding nature of neighboring metals towards oxygen resulted in either strong or weak adsorption energies relative to the pure system using the Co active site. The presence of Fe as a surface neighbor strongly attracts oxygen, causing an expansion of lateral Co-O bonds. This results in weak O* and OH* bonds towards the active site (Co). In contrast, when the surface neighbor is Mn, and the sub-surface neighbor is Cr, these Co-O bonds contract, resulting in weaker O* and OH* adsorption (**Supplementary Figs. 23a-f** and **24a-f**). The charge density difference analysis clearly shows that such expansion of Co-O bonds creates electron depletion along Co sites causing strong O* and OH* adsorption compared to pure and contraction cases (**Supplementary Figs. 23g-i** and **24g-i**). Finally, the equatorial expansion-contraction is also compensated by the axial sub-surface oxygen M-O bond to shorten-elongate, respectively (**Supplementary Figs. 25** and **26**).

Supplementary Fig. 23 | The O* binding energy and charge density plot for Co active site in HEO and pure system. The HEO system with strong O* binding energy in **a** pure spinel Co₃O₄ surface in **b** HEO system with weak O* binding energy. The strong binding resulted from the lateral expansion of Co-O bonds in **d** than the pure system in **e**; while weak binding resulted from lateral expansion in **f** of Co-O bonds. The charge density plots show that the local environment of active sites is mostly electron-deficient, resulting in strong O* binding in **g** than pure in **h** and a weak O* binding system in **i**. (iso-surface value 0.04 e/Å³. Cyan: charge depletion and yellow: charge accumulation)

Supplementary Fig. 24 | The OH* binding energy and charge density plot for Co active site in HEO and pure system. The HEO system with strong OH* binding energy in **a**, pure spinel Co₃O₄ surface in **b**, HEO system with weak OH* binding energy in **c**. The strong binding resulted from the lateral expansion of Co-O bonds in **d** than the pure system in **e**, while weak binding resulted from lateral expansion in **f** of Co-O bonds. The charge density plots show that the local environment of active sites is mostly electron-deficient, resulting in strong OH* binding in **g** than pure in **h** and a weak OH* binding system in **i**. (iso-surface value 0.04 e/Å³. Cyan: charge depletion and yellow: charge accumulation)

4) From the experiment it is concluded that some FeNi-OOH is formed. The calculations show that the overpotential of FeNi-OOH is higher than for the best HEO sites, however only 5% of the HEO sites have overpotentials below 0.4V, while the surface of FeNi-OOH have identical sites with an overpotential of 0.37V. Can you be sure that the observed activity stems from the HEO and not at least partly from FeNi-OOH?

Response: We know that the XPS peak intensities measure how much of an element is at the surface, which means that we can assume that the atomic rearrangement or new phase formation at the surface might occur during the electrolysis. Here, the increased intensities of trivalent states of Co, Fe, and Ni represents Co, Fe, and Ni can possibly be the cations in (oxy)hydroxide species. Also, the increased peak intensity of OH in the O 1s region may derive from the formation of (oxy)hydroxide species. For the future project, we also plan to conduct the elemental study using ex-situ, post-situ, and probably in-situ XAS as the next step to investigate the mechanism of atomic/structural rearrangement by the electrochemical measurement. We hope this can figure out the structural and elemental changes in each element before, after, and during the electrolysis.

Supplementary Fig. 18 | Ex-situ XPS elemental surveys show that the oxidation state of the HEO sample (CoFeNiCrMn) changes over time with more trivalent states and OH groups.

And as the reviewer questioned, it is true that some of the activity can originate from Fe-NiOOH. However, some of the oxides show no sign of Fe-NiOOH, but they still show very low overpotential of CoFeNiMn (379 mV), CoFe (418 mV), Co (433 mV), and Fe (522 mV). Therefore, we believe that HEO contributes to the activity significantly even though Fe-NiOOH was formed during the electrochemical measurement as the overpotential value was still lower than Fe-NiOOH.

Minor comments:

- Abstract, L27-28: Incorrect sentence

Response: We have revised the sentence.

Revision: (Page 1, Abstract)

“The rapid sol-flame method is successfully employed to synthesize spinel oxides with up to five 3d-transition metal cations (Co, Fe, Ni, Cr, and Mn) with different degrees of entropy.”

- Methods section: Some information is missing: Which values of U were employed and how was this chosen? How many layers are in the surface models?

Response: We used the U values from Material Projects. The U values for Cr, Mn, Fe, Co, and Ni are 3.5, 3.75, 4.3, 3.32, and 6.45 respectively. There are four (4) layers we used in our surface model where the bottom two layers are constrained.

- Methods section: What is meant by “5*5 permutation cells”?

Response: We have corrected this as “by using 5*5 permutation (${}_5P_5$)” in the method section of revised manuscript.

Reviewer #3 (Remarks to the Author):

In this paper, the authors reported the high entropy oxide (HEO) based on Co, Fe, Ni, Cr, and Mn prepared using a rapid sol-flame method. The authors use theoretical and experimental approaches to scrutinize the OER activity and stability for the obtained spinel-type HEOs. The manuscript can be reconsidered for publication after addressing the following comments.

Response: We appreciate the reviewer's positive opinion of our theoretical and experimental study on the HEO system. Below, we have attempted to answer all the raised comments by the referee.

1. The authors claim "The X-ray diffraction (XRD) of the resulting HEO in Fig. 2b shows a pure spinel Fd-3m (227) crystal structure with ICDD 00-01-0325." However, there is an obvious diffraction peak at around 33° , which does not belong to the Fd-3m (227) spinel crystal structure. Likewise, such an impurity peaks can be seen in Supplementary Fig. 5. This has undermined the discussion

Response: We appreciate the reviewer's comment. The diffraction peak at 33° represents alpha- Fe_2O_3 formed due to the presence of excess iron atoms or precursors in the mixture solution, which could not participate in the formation of HEO (See Fig. 1 in [ref 56]). However, even if Fe_2O_3 exists in our resultant, the overpotential of HEO (309 mV at 10 mA cm^{-2}) is much lower compared to the previously reported Fe_2O_3 anodes (440 mV at 10 mA cm^{-2} , See Table 1 in [ref 55] and 500 mV at 0.5 mA cm^{-2} , See Table 1 in [ref 54]). Therefore, we assume that the performance of HEO may be even higher if Fe_2O_3 does not exist. As the reviewer suggested, we marked the peak at 33° in Fig. 2b and described the existence of Fe_2O_3 impurity in HEO in the manuscript.

Revision: (Page 6)

"The tiny diffraction peak at 33° represents alpha- Fe_2O_3 probably formed due to the presence of excess iron precursors in the mixture solution, which could not participate in the formation of HEO.⁵⁴⁻⁵⁶"

Fig. 2 | Characterization methods for as-synthesized spinel HEO. **a** Schematic of the sol-flame synthesis process. **b** XRD and **c**, **d** high-resolution TEM images of HEO (inset: FFT image). **e** SAED pattern and **f** EDS-mapping images of Co, Fe, Ni, Cr, Mn, and O. **g** Dark-field image of HEO particles and **h** EEL spectra extracted from the four points with a distance of 1.5 nm between each point (0.5~5 nm) in **g**, showing O-K edge and L_{2,3}-edges of Co, Fe, Ni, Cr, and Mn. **i** Fe L₃/L₂ and Co L₃/L₂ white-line intensity ratio for the four points (0.5~5 nm) in **g**. **j** The energy difference between O K-edge pre-peak and main peak (energy separation, ΔE) at the four points (0.5~5 nm) in **g**.

2. In page 6 and 7, please double check the d-spacings for their corresponding diffraction planes carefully for both the synthesized sample and the reference NiFe₂O₄.

Response: We appreciate the reviewer's comment. We carefully double-checked and revised the d-spacing values for the synthesized particle and the NiFe₂O₄ reference.

Revision: (Page 7)

“In addition, the high-resolution TEM image (**Fig. 2d**) and the selected area electron diffraction (SAED) pattern (**Fig. 2e**) reveals that those nanoparticles are the polycrystalline phase of spinel oxide, mainly indexed to (220), (311), and (222), corresponding to the d-spacings of 0.290, 0.252, and 0.240 nm, respectively. The d-spacing values slightly deviate from the reference NiFe₂O₄ (00-010-0325), 0.2948, 0.2513, and 0.2408 nm for (220), (311), and (222), which is probably attributed to the atoms with different ionic radius in the lattice.”

3. The reduce peak intensities in the Cr 2p and Fe 2p XPS spectra with the catalysis time indicates the leaching of the metals during electrolysis. The authors are advised to provide data to show the metal leaching during OER, otherwise, please explain why the peak intensity is reduced.

Response: We have measured the ex-situ XPS elemental surveys to show the changes in oxidation states of each element Co, Fe, Ni, Cr, and Mn over time, as shown in **Supplementary Fig. 18**. We assume that there are some reasons for the decrease in the peak intensity.

First, as the reviewer said, Cr and Fe could be dissolved during the electrolysis [**ref 66**]. As the reviewer suggested, we investigated how much the constituent elements were dissolved to the electrolyte over time using inductively coupled plasma mass spectrometry (ICP-MS) in **Supplementary Fig. 19**. We found that mostly certain amount of Cr was definitely dissolved, and a little amount of Ni was also dissolved. Based on the certificate of analysis for the trace elements in water from NIST, quite a large amount of Fe is basically detected from pure water. For the structural change, we assume that the (oxy)hydroxide species are produced since the OH intensity increased a lot based on the XPS result and its cation can be Co, Fe, and Ni as their trivalent ratio increased, but mainly in Fe and Ni.

Supplementary Fig. 19 | Inductively coupled plasma mass spectrometry (ICP-MS) analysis. ICP-MS result for trace elements in the water reported by the National Institute of Standards &

Technology (NIST) and the dissolved elements in the 1 M KOH electrolyte after the electrochemical measurements (10 min, 1 hour, and 5 hours).

We know that the XPS peak intensities measure how much of an element is at the surface, which means that we can assume that the atomic rearrangement or new phase formation at the surface might occur during the electrolysis. Here, the increased intensities of Co and Ni represent Co and Ni atoms moved to the surface during the electrolysis, while vice versa for Fe and Cr. Third, the increased peak intensity of OH in the O 1s region may derive from the formation of oxyhydroxide or hydroxide species. We plan to conduct the elemental study using ex-situ, post-situ, and in-situ XAS for a future project. We hope this can figure out the structural and elemental changes in each element before, after, and during the electrolysis.

Supplementary Fig. 18 | Ex-situ XPS elemental surveys show that the oxidation state of the HEO sample (CoFeNiCrMn) changes over time with more trivalent states and OH groups.

Revision: (Page 11)

“In addition, we investigated how much the constituent elements were dissolved to the electrolyte over time using inductively coupled plasma mass spectrometry (ICP-MS) in **Supplementary Fig. 19**. We found that mostly certain amount of Cr was definitely dissolved which is well-known from the previous report,⁶⁶ and a little amount of Ni was also dissolved. Based on the certificate of

analysis for the trace elements in water from the National Institute of Standards & Technology (NIST),⁶⁷ quite a large amount of Fe is basically detected from pure water.”

4. Please spell “EC” that appears in “EC measurement” in page 10.

Response: We corrected the word from EC to Electrochemical.

5. The authors say that “... and divalent state in Mn increase within a couple of hours of the test.” However, visual inspection of the ex-situ XPS Mn 2p spectra (Supplementary Fig. 16) suggests positive slight of the spectra with the time. Please double check and clear this paradox.

Response: Thanks for the reviewer’s question. The relative amount of Mn²⁺ (divalent state) increases over time as shown in **Supplementary Fig. 18** in which the y-axis represents the trivalent to divalent cation ratio.

6. “Detailed XPS analysis in Supplementary Fig. 16 indicates that the Fe³⁺ and Ni³⁺ and OH concentrations were increased at the surface for 5 hours. This result demonstrates that the surface Fe-NiOOH was formed during the EC measurement, which influenced the slight change of overpotential (Fig. 3e). Although it is well known that the surface oxyhydroxide is also a promising OER catalyst, HEO still plays an important role as support contributing to the great stability.” The question is that why only Fe-NiOOH formed during the electrochemical measurement? What happened to the other metals? In other words, solid evidence and discussion must be provided to support the statement above.

Response: We agree with the reviewer’s concern. From the post-situ XPS result, all the elements (Co, Fe, Cr, and Ni) except for Mn made the trivalent state (Cr wants to stay in the trivalent state) as shown in **Supplementary Fig. 18**, which means those elements presumably make the OOH phase as OH peak intensity increases. However, there is a very slight change in Co³⁺ and no change in Cr³⁺. That is why we believe that Fe and Ni can be major contributors to forming an (oxy)hydroxide. As the reviewer said, we described the possibility that all the elements except for Mn can produce (oxy)hydroxide species based on our post-situ XPS result in the manuscript.

Revision: (Page 11)

“Detailed XPS analysis in **Supplementary Fig. 18** indicates that the Co³⁺, Fe³⁺, and Ni³⁺ and OH concentrations were increased at the surface for 5 hours, mainly in Fe³⁺ and Ni³⁺, in which this result demonstrates that the surface (M)OOH (M = Co, Fe, Ni, and/or Cr) was formed during the electrochemical measurement, which influenced the slight change of overpotential (**Fig. 3e**).”

7. Why does the existence of Mn lead to degraded performance?

Response: We have already discussed in the manuscript under “**the role of local strain on HEO activities**” section. The activity of HEO mostly depends on neighboring elements which are responsible for local expansion (or contraction) to equatorial Co-O bonds causing the stronger (or

weaker) O* and OH* binding. The presence of **Mn** as a surface neighbor (most stable position in octahedral site) weakly attracts lateral oxygens, causing a contraction of lateral Co-O bonds, which eventually results in weak O* and OH* binding energy and overall OER activity (**Supplementary Figs. 23 and 24**).

Supplementary Fig. 23 | The O* binding energy and charge density plot for Co active site in HEO and pure system. The HEO system with strong O* binding energy in **a**, pure spinel Co₃O₄ surface in **b**, HEO system with weak O* binding energy. The strong binding resulted from the lateral expansion of Co-O bonds in **d** than the pure system in **e**; while weak binding resulted from lateral expansion in **f** of Co-O bonds. The charge density plots show that the local environment of active sites is mostly electron-deficient, resulting in strong O* binding in **g** than pure in **h** and a weak O* binding system in **i**. (iso-surface value 0.04 e/Å³. Cyan: charge depletion and yellow: charge accumulation)

Supplementary Fig. 24 | The OH* binding energy and charge density plot for Co active site in HEO and pure system. The HEO system with strong OH* binding energy in **a** pure spinel Co₃O₄ surface in **b**, HEO system with weak OH* binding energy in **c**. The strong binding resulted from the lateral expansion of Co-O bonds in **d** than the pure system in **e**, while weak binding resulted from lateral expansion in **f** of Co-O bonds. The charge density plots show that the local environment of active sites is mostly electron-deficient, resulting in strong OH* binding in **g** than pure in **h** and a weak OH* binding system in **i**. (iso-surface value 0.04 e/Å³. Cyan: charge depletion and yellow: charge accumulation)

Reviewer #4 (Remarks to the Author):

This work deals with high entropy oxides (HEO) as OER catalysts. The work is an attempt to link DFT calculation with experiments. One of the main claims is that HEOs are better OER catalysts than mono, binary etc. catalysts. The reason would be that Co^{3+} is the active site, based on DFT and material characterizations.

HEOs are a very complex class of materials for which establishing a clear-cut structure and composition – OER performances is very challenging. The work is of quality but is nevertheless not compelling enough and it presents severe limitations as described below.

About the approach:

(i) the structural and microscopic characterizations are only conducted on the as-prepared HEO and other oxides. This is very annoying. Authors indeed conclude that there is more Co^{2+} and more Fe^{3+} close to the surface while they claim Co^{3+} is responsible for the OER activity. Authors must be aware that the surface region of most OER catalysts is known to undergo structural changes (see more below). Therefore, operando characterizations (for instance XRD, XAS are quite feasible) or, at least, presenting detailed post-operation characterizations appears indispensable.

(ii) the analysis of electrochemical data is superficial. Authors never address the key question of the normalization of the OER current while this is a prerequisite to compare different catalysts. Many works have addressed this difficult question and these works are not cited. Proper benchmarking of samples needs to be conducted.

(iii) the coupling between DFT and experiments appears very weak (see more details below). In addition, as stated above, the real OER active surface is probably not a (001) facet.

About the claims:

- HEO are claimed to be better performing than mono, binary etc. oxides. This conclusion is biased. Reality is much more complex than this. In fact, the performances of binary oxides depend on their composition and at optimum composition it may be better than that of corresponding mono element oxides. See literature about CoFe and NiFe oxides: In both cases, an optimum composition that departs from 50/50 enables reaching a better OER activity than the corresponding mono-oxides.

- DFT suggests that Co^{3+} sites are activated at low overpotential while a larger overpotential is required to activate Cr and Fe sites. In the first place, it is not convincing that the experimental situation (current $< 10 \text{ mA/cm}^2$) fits the conditions of low overpotential in DFT. Second, Co sites are in sufficient proportion at the surface to be characterized in operando. Last, XPS and EELS analysis suggest the present of Co^{2+} at the surface.

- Role of local strains at Co sites. Again, authors should provide experimental data in support of this statement. As said before, the structural characterizations are not sufficiently coupled with DFT. If strains are present at the surface of the grains, these should be detectable by operando XRD, because, after all, Co sites must occupy a good fraction of all surface sites.

Recommendation: In its present form the work is not recommended for publication, and it should be rejected. The experimental part is rather a routine applied material science study in relation with

OER. Authors should provide compelling experimental data that support that HEOs present better intrinsic OER activity. Relying on DFT is very dangerous since the calculation is performed on a perfect (001) plane while the real surface of the HEO likely undergoes a restructuring as many OER catalyst do. In addition, recent experimental and ab initio studies suggest that the OER active site may be composed of 2 atomic sites.

Response: HEO represents an exciting opportunity in the field of catalysts and materials research due to its vast tunability. For example, plenty of HEO-based works have been reported for OER activity, which shows comparable activity with RuO₂ or IrO₂ catalysts. [**Sustainable Energy Fuels**, 2022, 6, 1479-1488, **ACS Appl. Energy Mater.** 2022, 5, 8, 9292–9296, **Journal of Alloys and Compounds** 868 (2021) 159064, **Front. Energy Res.** 2022 10:942314]

However, it is a great challenge to study and under the catalytic properties of HEO both experimentally and theoretically. Our work is a starting point to show how we can approach this problem with DFT and experiments. We hope that our work will inspire others to continue to improve and innovate new methods to study HEO.

In addition, spinel oxides are well known to produce cubic nanoparticles with exposed (100) facets even when under high-temperature synthesis. There are several studies already reported that (100) facet has more stability phase than others. [**Phys. Chem. Chem. Phys.**, 2021, 23, 23768-23777, **ChemCatChem** 2011, 3, 1159 – 1165, **Surface Science** 677 (2018) 278–283, **Nanotechnology**, 2012, 23, 325703]

Our XRD results show that HEO remains in spinel structure, as seen from **Fig. 2b-2d**, and **Supplementary Figs. 5 and 6** (XRD and TEM images). To comment on the “The 2 atomic sites OER approach”, reviewer likely refers to dual-site OER mechanism. Dual site mechanism is quite complicated and has a very narrow window of operation when it’s preferred over single site mechanism (both sites need near optimal OH*, O* and low desorption of O₂). When searching for the optimal energetics of OH* and O*, these apply to both, dual and single site OER mechanism. Single site mechanism offered here is just the most general first level approach applied to 3d-metal HEOs for the first time.

Specific comments:

1. Experimentally, the synthesis of HEOs relies on rapid thermal quenching of a melt. Should not it be expected that the structure is metastable? On which basis is calculated the data in Supplementary Fig. 2 which gives the crystal field energy of materials. Is the calculation considering a structure at equilibrium?

Response: Rapid thermal annealing and quenching of a melt result in bulk high-entropy oxides close to equilibrium at the ambient environment

(**Figure 1** in [**Adv. Energy Mater.**2022, 12, 2200742] and [**Journal of the European Ceramic Society** 40 (2020) 1644–1650]), which is a spinel phase in a normal or inverse structure. This is the main reason why we used an idealized spinel structure for our study. We have used the crystal field energy (<https://doi.org/10.1017/CBO9780511524899>) from the reference.

2. It is extremely difficult to understand how the underlying 2D volcano plot in Fig. 4a has been constructed.

Response: The details of the volcano plots constructions are very simply summarized in our original 2013 JACS paper (10.1021/ja405997). Here, we consider three active sites for this study e.g., Co, Fe, and Cr. For each active site, we calculated O*, and OH* adsorption energy and used fitted OOH* scaling relationship (Supplementary Fig. 20) to get theoretical overpotentials using single site OER mechanism (Supplementary Table 5).

Fig. 4 | OER site-dependent activity of the equimolar HEO. **a** OER activity volcano plot as a 2D heat map of overpotentials based on binding energy calculation of O* and OH*, and scaled OOH* values for HEO systems, grouped by the active site along with the pure spinel systems. **b** Active sites (Co, Cr, Fe) distribution as a function of OER overpotential for HEO systems. **c** Predicted activity trends for different spinel oxide systems with explicitly calculated OOH*.

Revision: (Page 17)

“Using these descriptor values, for each surface, we have calculated O-OH*, and OH* energies and place them on the OER 2D map which is constructed from the function of Max (OH, O-OH, OH+1.23). Each active site has 120 unique combinations and overpotentials. The colors bar in the heatmap indicates the region on each overpotential.”

3. The graph in Fig. 4b is confusing. Authors must better explain why only a fraction of sites are activated at a given overpotential. This must have some relation with Figs. S17 and S18. In addition, it is recommended plotting, for each element the % of activated sites (normalized to the TOTAL number of surface sites of this element). A vertical log scale would be useful. With one graph per element, one could catch which type of surface site is most active at a given overpotential.

Response: Thank you for your suggestion. **Fig. 4b** is actually a statistic from **Fig. 4a**, which indicates how many particular sites fall under certain applied potential among all site combinations. The figure shows approximately 5% of the overall combinations (360 sites) are activated under a small overpotential of less than 0.4 V. Within 5% of sites, the maximum (around 70%) activity is coming from Co active site. Along with **Supplementary Figs. 21 and 22**, in **Supplementary Figs. 23 and 24**, we have shown the origin of high activity. The minimum overpotential containing sites show optimum binding strength to O*, and OH* adsorbates resulting in very good OER activity.

Revision: We have revised Fig. 4b and made it simpler (see the answer in comment #2 of the same reviewer).

4. The discussion of the role of local strain on activity of Co sites is not very convincing in the present form (bottom of page 13). Authors should provide experimental data in support of this statement. If strains are present at the surface of the grains, these should be detectable by operando XRD or XANES, because, after all, Co sites represent a good fraction of all surface sites.

Response: Fundamentally, the lattice structure of the multi-element solid solution phase (of HEO) is distorted, which leads to microstrain. To provide experimental evidence, we measured the XRD for some of the spinel oxides including HEO and used these XRD patterns to obtain the Williamson-Hall plot, which is used to calculate the crystallite size and microstrain of each spinel oxide (**Supplementary Fig. 27 and Supplementary Table 6**). Here, we used the Scherrer equation to estimate the grain size. Here, we used the Scherrer equation and Williamson-Hall plot to estimate the grain size and microstrain,

$$D = \frac{K\lambda}{\beta \cos \theta}$$

, where D is grain size, K = 0.9 (Scherrer constant), $\lambda = 0.15406$ nm (wavelength of the x-ray sources (Cu K α)), β = FWHM (in radians), and θ = peak position (in radians).

In the Williamson-Hall plot,

$$\beta \cos \theta = 4\varepsilon \sin \theta + \frac{K\lambda}{D}$$

, where ε = microstrain.

As shown in **Supplementary Fig. 27**, HEO shows the highest microstrain among different spinel oxides. We added this in the Supplementary Information and described it in the manuscript.

Revision: (Page 15)

“In order to provide experimental evidence of the microstrain, we additionally measured XRD for some of the spinel oxides including HEO and employed the XRD patterns to obtain the Williamson-Hall plot, which can be used to calculate the microstrain and crystallite size of each spinel oxide. As shown in **Supplementary Fig. 27 and Supplementary Table 6**, HEO shows the highest microstrain among the various spinel oxides. We assume that, fundamentally, the lattice structure of the multi-element solid solution phase (of HEO) is distorted due to a variety of elemental bonds,

which leads to microstrain and is in good agreement with the calculation of the bonding and prediction of HEO activity.”

Supplementary Fig. 27 | Estimation of microstrain and crystallite size from XRD. **a** XRD result for several spinel oxides and **b** calculated crystallite size and microstrain for each oxide. Here, we used the Scherrer equation and Williamson-Hall plot to estimate the grain size and microstrain,

$$D = \frac{K\lambda}{\beta \cos \theta}$$

, where D is grain size, K = 0.9 (Scherrer constant), $\lambda = 0.15406$ nm (the wavelength of the x-ray sources (Cu K α), β = FWHM (in radians), and θ = peak position (in radians).

In the Williamson-Hall plot,

$$\beta \cos \theta = 4\epsilon \sin \theta + \frac{K\lambda}{D} \cos \theta$$

, where ϵ = microstrain.

Supplementary Table 6 | Crystallite size and microstrain for different spinel oxides calculated from XRD results in Supplementary Fig. 27.

	Crystallite Size (nm)	Microstrain
NiFe	42.47	1.03
CoCr	17.26	1.46
CoNiFe	41.49	1.55
CoNiCr	9.64	1.4
CoNiFeCr	12.48	3.03
CoNiFeCrMn	24.02	8.25

5. Supplementary Fig. 4: It is very difficult to catch the message from this figure. A complete description (in SI) is mandatory to explain this figure.

Response: **Supplementary Fig. 4** indicates the stability comparison of various types of spinel oxide mixing together. We have discussed this briefly in the supplementary information.

Revision: We have added a discussion in **Supplementary Fig. 3**.

Supplementary Fig. 3 | Energy stability for the ternary elemental mixture for HEOs. We mix three (3) spinel structures together and calculate binding/formation energy relative to the sum of each stable bulk structure (Fe_3O_4 Inverse, Co_3O_4 & Ni_3O_4 Normal). The B combination where we calculate individual bulk energy of Fe_3O_4 Normal, Co_3O_4 & Ni_3O_4 Inverse, and the summation energy show less stable as compared to the reference. On the other hand, C and D combinations show mixture systems where elements within the same (C) and different (D) phase swap their position. From the energy calculation, we can see that the C combination possesses the most negative energy, indicating the most stable ternary mixture.

6. The average thickness of layers is not mentioned. What is the “desired thickness” in page 18?

Response: We appreciate the reviewer’s comment. We measured cross-sectional SEM images to measure the average thickness. The thickness of the HEO film can be measured as around 50~80 nm depending on the roughness of the FTO/glass substrate as presented in **Supplementary Fig. 5**.

Revision: We have added this information to **Supplementary Information**.

Supplementary Fig. 5 | SEM and TEM images of the HEO sample. a HEO particle on FTO/glass substrate. **b** Higher magnification SEM image of HEO particle. **c** Cross-section images of HEO film on FTO/glass substrate. **d** SEM-EDS mapping images showing all elements included. **e** Detailed TEM images.

7. XRD: page 5, line 161 and Supplementary Fig. 5: the term “in situ” refers to temperature dependent XRD measurements. There might be a confusing with in situ (i.e. in electrochemical environment) XRD. Authors should determine the grain size from their data and compare it with TEM images. Showing similar data for mono, binary etc. oxides is mandatory to compare the ECSA (see comments below).

Response: We got the reviewer’s point. Some people study the behavior of materials and the chemical and structural changes during operation or in different environments, which is in-situ measurement. In-situ varies depending on what people use. Some people study temperature dependence, while others do in their operating systems. Here, we changed the term in-situ to in-situ temperature-dependent XRD. Based on the XRD result in **Supplementary Fig. 27**, we could calculate the crystallite sizes of different spinel oxides, and their values are shown in **Supplementary Table 6**. Also, we have measured the ECSA to compare with the grain sizes (**Supplementary Fig. 14**). However, we could not find a similar tendency between ECSA and the crystallite size from XRD. This is probably due to the different roughness of the FTO/glass substrate.

Supplementary Fig. 14 | Surface area analysis. **a** Electrochemical surface area (ECSA) and **b** the normalized ECSA for the different spinel oxides by ECSA of FTO/glass substrate.

8. XANES: Page 7, lines 190-195 and Supplementary Fig. 10: the analysis is too superficial. Considering the slope of the XANES plot is definitely not appropriate. The edge energy is the relevant parameter and it must be quantitatively measured and converted into an average oxidation state using the calibration plot. The calibration plot must be presented and authors must give values (with error bar) of the Edge energies and corresponding average oxidation state of elements. Authors perfectly know that XAS gives the average oxidation state of elements. Therefore, additional characterizations and analysis are required to gain insights into the valence state of cations at the surface.

Response: We appreciate your meaningful comments on the XAS result. We also agree with the reliability of the comparison of the XANES plot, thus we put the quantitative comparison of oxidation states using the calibration curve as seen in **Supplementary Fig. 10c-d**. The calibration curve was calculated by the integrated method using the lower (0.3) and upper limit (1.0) from the XANES slope. Also, the error bars are the standard error of the regression slope for the uncertainty in oxidation states. Both CoFeNiCr and HEO are located between Co(II) and Co(II, III) for Co, and located between Fe (II, III) and Fe (III). Thus, we could estimate the oxidation states of Co (~2.13) and Fe (~2.80) for CoFeNiCr and Co (~2.20) and Fe (~2.70) for HEO.

As the reviewer said, soft-XAS may be employed to distinguish the oxidation state between surface and bulk, using total electron yield (TEY) and fluorescence yield (FY) modes. TEY mode can detect the oxidation state in the outer ~5–10 nm of the particle and FY mode probes deeper into the bulk of the material ~50–100 nm. Thus, the difference between TEY mode (surface) and FY mode (sub-surface/bulk) or hard-XAS (bulk) can provide information on the difference between the surface and bulk oxidation states of the particles. However, this time we could not measure the soft-XAS due to the restricted access. We may use it to figure out the surface oxidation state of each element for a future project. [**Nano-Micro Lett.** (2019) 11:47]

Supplementary Fig. 10 | Normalized X-ray absorption near-edge structure (XANES) analysis for HEO with other reference oxides. a XANES spectra of the Co K-edge. The inset figure shows the expanded section from 7718 to 7725 eV. The references include CoO (Co (II)) and Co₃O₄ (Co (II, III)). **b** Fe K-edge XANES spectra. The inset indicates the expanded region from 7123 to 7127 eV. The reference oxides are Fe₂O₃ (Fe (II, III)) and Fe₃O₄ (Fe (III)). **The calculated c Co and d Fe oxidation states from the XANES spectra.**

Bulk transition metal Co and Fe oxidation state measured by transmission-based TM K-edge X-ray absorption spectroscopy (XAS). First, both Co and Fe oxidation states were estimated by comparing Half E0 edge height and using the previously reported integral method, which was found to be a better descriptor of the transition metal oxidation state rather than the half-height or second derivative method. Here, the calibration curves were made using the integral method with lower and upper limits of 0.3 and 1. The slopes are the only values taken from these plots for comparing each oxidation state. The reference oxides are also listed to show the correlation between the excitation energy and metal valence.

Revision: (Page 7-8)

“Furthermore, the X-ray absorption spectroscopy (XAS) measurements (**Supplementary Fig. 10**) show that the bulk valence states of Co and Fe in HEO contain divalent and trivalent states. Based on the x-ray absorption near-edge structure (XANES) profiles, the quantitative comparison of oxidation state from the Co K-edge spectra (**Supplementary Figs. 10c and 10a**) shows the valence values of Co for both CoFeNiCr and HEO are located between Co²⁺ and Co^{2+/3+}. In addition, the higher Co valence state appears with adding Mn cation to CoFeNiCr, which is consistent with the XPS result. Moreover, the Fe K-edge of CoFeNiCr and HEO, and quantitative comparison of valence state (**Supplementary Figs. 10b and 10d**) are in the lower region than the reference materials. This implies that the relative Fe oxidation state of HEO is the lowest and tends to appear more divalent state. Thus, HEO exhibits the higher Co³⁺ than Co²⁺ and Fe²⁺ than Fe³⁺ throughout

the particle, compared to those of the lower entropy oxide (CoFeNiCr). A detailed experimental description of XANES is also shown in **Supplementary Fig. 10.**”

9. EELS: Fig. 2g-k and related text, page 7: Was EELS at different locations performed on other particles (how many?) to make sure that this is representative. In (h) the signal is rather noisy and one wonders about the precision of the integration of lines L2 and L3 (especially for Fe). All this should be better documented. Error bars are also requested. In Fig. 2i-j one would like to see plots of the intensity of individual lines L3 and L2 of all elements (with error bars) and not just the ratio L3/L2 for Fe and Co. Authors conclude that there is more Co²⁺ and more Fe³⁺ close to the surface while they claim Co³⁺ is responsible for the OER activity. Explain the discrepancy. This points to the need of operando spectroscopic characterizations.

Response: We agree with the reviewer’s comment and have remeasured EELS with a much higher resolution using a better TEM (the details will be described in the **Methods section**) for the five different particles to make an error bar. The new EEL spectra were taken every 1.5 nm from the surface to the bulk of each particle. Each raw EELS result with the corresponding ADF images was shown in **Supplementary Fig. 11**, which shows the data point with a consistent tendency. We used these data points from the five samples to make an error bar.

Based on these spectra, we could recalculate the L3/L2 ratio for Fe, Co, and O regions (**Fig. 2i and 2j**). Now, the more accurate measurements show that the L3/L2 ratio of Fe is lower at the surface, which means a lower oxidation state at the surface [**ref 57 and ref 60**]. Moreover, the L3/L2 ratio of Co is also lower at the surface, which represents the higher oxidation state at the surface. This more accurate result shows that the amount of Co³⁺ is higher at the surface (when we define the surface as below 1 nm.), which is an active site in HEO [**ref 58 and ref 59**]. For the O K region, the energy separation values are very diverse, so different oxidation states of five elements are mixed from the surface to the bulk [**ref 61**].

As the reviewer suggested, we are planning to measure operando XAS for a future project. We hope we can figure out the changes in oxidation states of each element in HEO during the EC measurement.

Fig. 2 | Characterization methods for as-synthesized spinel HEO. **a** Schematic of the sol-flame synthesis process. **b** XRD and **c**, **d** high-resolution TEM images of HEO (inset: FFT image). **e** SAED pattern and **f** EDS-mapping images of Co, Fe, Ni, Cr, Mn, and O. **g** Dark-field image of HEO particles and **h** EEL spectra extracted from the four points with a distance of 1.5 nm between each point (0.5~5 nm) in **g**, showing O-K edge and L_{2,3}-edges of Co, Fe, Ni, Cr, and Mn. **i** Fe L₃/L₂ and Co L₃/L₂ white-line intensity ratio for the four points (0.5~5 nm) in **g**. **j** The energy difference between O K-edge pre-peak and main peak (energy separation, ΔE) at the four points (0.5~5 nm) in **g**.

Supplementary Fig. 11 | Several spots of dark-field images of HEO particles and corresponding EEL spectra extracted from the points (3 to 4 points) with a distance of 1.5 nm between each point. The last two plots show the summarized Fe and Co L₃/L₂ ratio in which the same color of the dots are from the same particle. The average and standard deviation values are shown in **Fig. 3i**.

Revision: (Page 8)

“The electron energy loss spectroscopy (EELS) analysis (**Fig. 2g to 2j**) was performed on the HEO particle to quantify its chemical composition. The EEL spectra (**Fig. 2h**) show the presence of the L_{2,3}-edges of all five metal elements (Cr, Mn, Fe, Co, Ni) at four positions in **Fig. 2g** with a distance of 1.5 nm. Moreover, **Figs. 2i** plots the L₃/L₂ white-line intensity ratio for Co and Fe at those six positions. As the L₃/L₂ white-line intensity ratio was reported to be closely associated with the orbital occupancy of the 3*d*-transition metal elements and their oxidation states,^{55–58} the results suggest that the HEO has more Fe²⁺ and Co³⁺ on the surface and more Fe³⁺ and Co²⁺ toward the center of the HEO particle. Here, the error bars come from several spots of the different particles, and each value is shown in **Supplementary Fig. 11**. **Fig. 2j** shows the energy separation between O K-edge pre-peak to the main peak, and its larger values near the surface indicate the higher oxidation states of overall elements on the surface in which HEO has the random energy separation value at the four positions.⁵⁹ This is probably from the different oxidation states of five elements are mixed from the surface to the bulk.”

10. Page 7, line 171: XPS probes the near surface composition. You cannot state it gives the surface composition. For instance, it is impossible from the data to state that Co³⁺ or Co²⁺ is that the

surface and would act as active site. A reader would like to have the information in Table S2 closer to the main text.

Response: We thank the reviewer for this suggestion. Our XPS (PHI VersaProbe 3) provides the surface region of the first 1-30 monolayers of materials, which is sensitive to the top 5 nm of a sample. We know that all these layers could not act as active sites, but we can assume that the active site elements are dominantly distributed if the elements/oxidation states are dominant at the surface region which XPS detects. We believe that many researchers use XPS for the compositional and chemical analysis at the surface in which they show the ratio of the oxidation states of each element and connect it to the explanation for the active sites. We referred to some of the references. [ref 39, *Adv. Funct. Mater.* 2021, 31, 2106229]

We also described more about **Supplementary Table 2** in the manuscript. And based on the table, we got the $\text{Co}^{3+}/\text{Co}^{2+}$ ratio of each sample (**Supplementary Fig. 9**), showing the trend that when adding more elements, more Co^{3+} was generated, which acts as an active site for oxygen evolution reaction.

Revision: We have modified **Supplementary Table 2** and included **Supplementary Fig. 9**.

Supplementary Table 2 | The elemental concentration of each oxidation state of each element for multi-element spinel oxides from the XPS survey analysis.

	Co^{2+}	Co^{3+}	Fe^{2+}	Fe^{3+}	Ni^{2+}	Ni^{3+}	Cr^{3+}	Cr^{6+}	Mn^{2+}	Mn^{3+}	O
Co	0.158 ± 0.032	0.202 ± 0.045									0.640 ± 0.016
Fe			0.236 ± 0.043	0.100 ± 0.011							0.663 ± 0.053
CoFe	0.058 ± 0.0003	0.071 ± 0.001	0.145 ± 0.039	0.100 ± 0.032							0.626 ± 0.008
CoFeNi	0.042 ± 0.012	0.059 ± 0.021	0.032 ± 0.013	0.079 ± 0.024	0.088 ± 0.008	0.018 ± 0.016					0.683 ± 0.051
CoFeNiMn	0.038 ± 0.003	0.060 ± 0.015	0.030 ± 0.008	0.086 ± 0.030	0.091 ± 0.020	0.008 ± 0.005			0.024 ± 0.012	0.068 ± 0.009	0.593 ± 0.051
CoFeNiCr	0.039 ± 0.009	0.065 ± 0.005	0.027 ± 0.003	0.081 ± 0.005	0.081 ± 0.006	0.018 ± 0.008	0.084 ± 0.011	0.021 ± 0.002			0.582 ± 0.026
CoFeNiCrMn	0.034 ± 0.007	0.056 ± 0.005	0.059 ± 0.016	0.035 ± 0.011	0.088 ± 0.008	0.005 ± 0.002	0.066 ± 0.011	0.022 ± 0.003	0.014 ± 0.003	0.060 ± 0.011	0.561 ± 0.025

Supplementary Fig. 9 | $\text{Co}^{3+}/\text{Co}^{2+}$ ratio based on XPS from Supplementary Table 2.

11. Page 7, lines 183 - 185: “This implies that HEO has more Fe^{3+} and Co^{2+} at the surface, elaborated by electron energy loss spectroscopy (EELS) later”. The reasoning leading to this statement is obscure and the formulation looks strange. Once again, the claim in abstract is that Co^{3+} are the active site. Explain the discrepancy.

Response: The same answer as in the EELS comment #9.

12. Figs. S8 and S9: Obviously, the decomposition of spectra is not unique, which implies the determination of the relative concentrations of valence state per element cannot be given with the precision given in Table S2. Provide error bars. In addition, authors should give the atomic ratio of the different elements to judge whether there is a difference between the surface and the bulk composition.

Response: We agree that the reviewer’s suggestion. Thus, we measured more spots on the samples and made the averages/error bars to precisely compare the atomic ratio for each element for each sample (**Supplementary Table 2**). Also, we carefully deconvoluted the XPS spectra again. As we mentioned above, the sensitivity of our XPS is nearly the first 1-30 monolayers of materials, top 5 nm of a sample. Along with TEM-EDS, XPS shows near-equimolar concentration for at least the top 5 nm.

Revision: We have added Supplementary Figs. 7 and 8.

Supplementary Fig. 7 | Detailed XPS elemental survey of HEO particles. **a**, Cr 2p, **b**, Mn 2p, **c**, Fe 2p, **d**, Co 2p, **e**, Ni 2p, and **f**, O 1s. Cr has only a single oxidation state (Cr^{3+}) with peaks between 575.7 and 578.9 eV in the Cr 2p_{3/2} region. The Mn 2p_{3/2} region consists of the peak at 640.1 eV corresponding to the divalent state, while the peak at 642.2 eV belongs to the trivalent state. In the Fe 2p_{3/2} region, the peak at 709.7 eV indicates a divalent state, while the peak at 711.2 eV shows the existence of the trivalent state. The Co 2p_{3/2} spectra exhibit the Co^{3+} peak at 779.6 eV and Co^{2+} peak at 781.6 eV. Lastly, the Ni 2p_{3/2} has the dominant Ni^{2+} peak at 854.9 eV and the inferior Ni^{3+} peak at 856.5 eV. The valence state positions of all elements show a little deviation from the literature values as the HEO particle incorporates random bonding distances between the atoms.

Supplementary Fig. 8 | XPS elemental survey result of Fe 2p and Co 2p for different spinel oxides. **a** Fe 2p and **b** Co 2p to compare the amount of each oxidation state for multi-element spinel oxides.

13. It is very annoying that the authors do not make any attempt to normalize the current to a sensible parameter such as the electrochemical active surface area (ECSA). A normalization of the OER current by the geometrical surface area of the substrate may be fine for application oriented approach but it tells nothing about the intrinsic OER activity because the samples present a complex morphology and are perhaps porous. Not only, the electrochemically active surface area (ECSA) may be quite larger than the geometrical area but it may also vary from sample to sample. The set of data does not give enough information pertaining that the ECSA may be similar for all mono, binary ... HE oxides used in this study. A change in ECSA by a factor 3 is, for instance, enough to modify the overpotential by 15 mV to 30 mV, according to the Tafel slope. Table S4. This Table cannot be used to state that HEO are intrinsically better OER catalysts. A true benchmarking analysis is missing.

Response: As the reviewer suggested, we measured the ECSA of the different spinel oxides including HEO and normalized by ECSA of the FTO substrate. Here we found that HEO has a relatively smaller ECSA than other spinel oxides. Nevertheless, HEO has the highest activity toward OER. We can assume that if HEO has a larger surface area, it should have much higher

activity. However, we are not interested in the change of morphology and surface area for this project. We would want to make the structure simpler to easily compare the performance. That is why we fabricated the thin film on the FTO/glass conductive substrate and used the geometric area (0.6 cm^2). We know that many of the literature use powder samples loaded on the glass carbon electrode and they calculate the ECSA to normalize the area, but we assume that FTO can be considered flat as seen in the ESCA plot below.

Revision: We added the ECSA result in the Supplementary Information. (Supplementary Fig. 14)

14. The Tafel slope of CoO appears larger than previously reported for Co₃O₄.

Response: It could be possible however we did not provide any comparison of the Tafel slope between CoO and Co₃O₄.

15. Fig. 3c: this graph is rather misleading since a mono cation oxide (e.g. CoO) may be performing better than those reported here. In addition, binary oxides may perform better than mono oxides after optimization of the composition. The composition of different oxides should be given in the figure.

Response: As the reviewer said, mono oxide with different compositions in other literature may perform better than HEO in our project, as they may be optimized by composition control, morphology control, etc. However, our goal is not to optimize a particular catalyst, but to focus on the composition effect on the catalytic activity to explore the potential of HEO. Hence, we only fabricated thin film with equimolar concentration to easily and simply compare the performance of different entropy oxides from binary to senary.

16. The equivalent electrical scheme used to fit EIS must be justified. In addition, fitting procedure must be described because with so many parameters it is hard to believe that one can unambiguously determine R₂, the bulk resistance, by adjusting one spectrum. Why authors do not exploit the other parameters derived from the EIS modelling?

Response: In our work, the impedance raw data were simulated in terms of the circuit model known as the Armstrong-Henderson equivalent circuit. In this equivalent circuit, R₁ represents the uncompensated solution resistance, R₂ is the charge transfer resistance, and the resistance R₃ represents the parameter associated with the relaxation of the surface coverage of an adsorbed intermediate in the OER mechanism. As the reviewer suggested, we added the proper references to the manuscript.

Revision: (Page 19, Electrochemical measurement section)

“The EIS raw data were simulated in terms of the equivalent circuit model known as Armstrong-Henderson equivalent circuit, depicted in **Supplementary Fig. 13**.^{62–64}”

17. Page 1, line 29: “Such catalyst exhibits the highest OER activity of 309 mV at 10 mA cm⁻²”: this is a laboratory language, certainly not a scientific one. Please reformulate.

Response: We corrected the sentence. Thanks for the suggestion.

Revision: (Page 1, Abstract)

“Particularly, high-entropy oxide with five elements displays the superior OER activity and prolonged durability under alkaline conditions compared to lower entropy oxides despite partial surface oxidations.”

REVIEWER COMMENTS

Reviewer #1 (Remarks to the Author):

I have read a few times this revised manuscript, and compare it with previous version, together considering the reply to comments. In my opinion, the authors have quite well addressed referees' comment and largely improved the quality of the manuscript. I believe this is a nice work demonstrating synergistic effects of mixing and strain in high-entropy spinel oxides towards OER, and thus have no more question on the publication of the work in Nat. Communication.

Reviewer #2 (Remarks to the Author):

I thank the authors for their reply which has clarified my primary concerns. From this I understand that the assumption made in the calculations is that the 5 nearest neighbours to the active site consist of exactly one of each of the elements Fe,Co,Cr,Mn and Ni. This is a reasonable way to reduce the number of computational structures, and as such I am satisfied that the computational setup is sensible. Nevertheless, this assumption should be expressed clearly in the manuscript which is not currently the case, i.e. the sentence p4, l116 should be modified (suggested changes in parentheses):

In this setting, each active site at the top-most layer is surrounded by the five nearest elements (neighbours) ($< 3.5 \text{ \AA}$) (A-E, Fig. 1a), and these five neighboring sites (are assumed to be occupied by exactly one atom of each of the elements Cr, Co, Mn, Ni and Fe in any arrangement, ...) can be occupied by Cr, Co, Mn, Ni, and Fe in any arrangements, which leads to 120 distinct permutations per active site.

Furthermore, regarding the reply to question 3:

It is shown that strain leads to the change in adsorption energy on Co. The unit cell is quite small, and on a real surface the strain from a Co with neighbouring Fe might be relieved by the strain from a nearby Co with Mn neighbours. Did the authors investigate if the strain and resulting effects are the same in a larger unit cell with dissimilar active sites?

Reviewer #4 (Remarks to the Author):

See also attached report.

- What are the noteworthy results?

Yes because the experimental increase of OER activity by mixing more elements in HEOs seems robust. The interpretation of this result is however more dubious.

- Will the work be of significance to the field and related fields? - How does it compare to the established literature? If the work is not original, please provide relevant references.

Yes, provided the conclusions / interpretation are better supported by experimental data. This is partially the case. Comparison experiment / DFT is not convincing for systems at the heart of the discussion.

- Does the work support the conclusions and claims, or is additional evidence needed?

In part. Additional electrochemical characterizations are necessary to make sure that effects are significant above for HEOs with more than 4 elements. The question of microstrains need also to be more documented to establish correlation OER activity - microstrains.

- Are there any flaws in the data analysis, interpretation and conclusions? Do these prohibit publication or require revision?

Additional data are necessary. See attached report.

- Is the methodology sound? Does the work meet the expected standards in your field?

In part, operando characterization would be very useful.

- Is there enough detail provided in the methods for the work to be reproduced?

In part.

Revised Manuscript NatComm_379092_0

This work about HEO catalysts is certainly interesting and seems solid. In the present form, authors convincingly show that mixing more elements improves the OER activity of the HEO but more care is requested to discuss differences between the OER activity of 4-5 elements HEOs. The current interpretation of this result remains fragile and, it needs to be said that the lack of operando XAS or operando XPS characterizations is highly regrettable. Operando structural characterization would be also desirable to check possible modifications of the catalyst and make sure that the surface is indeed as authors assume.

The revised paper needs further major revision.

Main comments:

1) About the dependence of OER activity as a function of number of elements within HEO.

A global trend seems obvious from the series of LSV. The OER activity increases by more elements. However, above 4 elements the differences between the CVs fall almost within the experimental reproducibility of the OER response. In particular, the overpotential difference is only 18 mV between CoFeNiCr and CoFeNiCrMn catalysts, which are the two systems at the heart of the discussion in this manuscript. Authors perfectly know that the CVs of independent samples of same composition / structure may shift by several 10 mV. Therefore, authors should convince the reader that they are discussing real effects.

Recommendation: authors must provide additional electrochemical data to show that the dispersion of CVs is small compared to the 18mV discussed above. They must show the CVs of several samples (in particular those mentioned above) prepared with the same composition. This will allow estimating an error bar for overpotentials, a parameter that is crucially missing.

2) Are mixed Co³⁺ sites the most OER active sites of the HEO?

The link between authors' statement and experimental characterizations remains fragile in absence of operando measurements. There is also a real problem when comparing DFT data with experiments. In Fig. 4c the predicted OER overpotential is 0.2 V larger for CoFeNiCr (0.49 V) than for CoFeNiCrMn (0.29 V) while these two catalysts present a rather similar OER activity (see above). This huge discrepancy here between DFT and experiments needs to be commented for it somewhat puts in question the whole reasoning.

3) About the role of microstrains on OER activity.

The data in Table S6 indeed show that strains increase by mixing more elements (btw, what this the reference crystal structure?). Nevertheless, these data do not prove that the micro strains are the cause for the enhanced OER activity.

Recommendation: To demonstrate their point, authors should show correlation OER activity – strains over the widest possible set of samples. They could measure the micro strains for all 4 to 5 elements HEOs they have synthesized and check whether they can establish correlation strain - activity.

Other comments:

1) XAS:

- Panels (c-d) in Fig. S10 are confusing: These plots are supposed to be calibration plots and authors only show 2 data points corresponding to CoNiFeCr and CoNiFeMnCr catalysts and one straight line.

This leaves the impression that the straight line is a “fit”, which would be meaningless with only 2 data points. Please show the data points corresponding to reference samples with their error bars. Give the value of slope and compare it with existing literature.

- In caption of Fig. S10, Fe₂O₃ is not Fe(III).

- Bulk oxidation states of elements should be compared with DFT values (this information is not found).

2) XRD.

- The reader wonders how XRD is experimentally combined with the flame annealing technique. Experimental details should be given.

- Fig. S4 : the spectrum at 25°C measured is poorly commented. Was it measured after the sample has cooled down after annealing at 1000°C? Please specify. Bragg peaks must be indexed.

- Please show the Williamson-Hall plots and give error bars for the microstrains.

- It would be interesting to give the lattice parameters of the series of HEO.

3) EIS:

- The equivalent circuit given in Fig. S13b was originally proposed by Conway in Ref. 62 (with capacitance instead of CPE) and was also used in ref. 64 (see right scheme) but not in ref. 63.

Conway used this scheme to discuss the mechanisms of HER accounting for the relaxation of *surface coverage of an adsorbed intermediate* (left figure): R_{ct} is the usual charge transfer resistance, which accounts for the kinetics of the reaction; The parallel (R_p/C_p) circuit accounts for *the relaxation of the surface coverage of an adsorbed intermediate in the HER mechanism* and the CPE accounts for the dispersion of the Cdl. It seems legitimate to apply it for OER for reaction intermediates are also adsorbed during the reaction.

Authors' use the same circuit but interpret it quite different. In fact, R_2 (analogue of R_{ct} in left scheme) becomes the charge resistance inside *bulk* catalyst, whatever this means. The parallel R_3/CPE circuit (analogue of R_p/C_p) becomes the charge transfer resistance between catalysts and electrolyte. This interpretation is not physically sound.

Authors must explain / justify their circuit and should not refer to Refs. 62-64 for their interpretation is different. Authors test their model. In particular, by discussing the two CPE, provided the factor “n” (not given) is close to 1 (otherwise the model is not suitable). The estimated Cdl could be compared with that used to estimate the ECSA.

4) XPS:

- The increase of the ratio Co³⁺/Co²⁺ after OER (Fig. S18) was to be expected because OER corresponds to strongly oxidizing conditions. This also happens at the surface of Co spinel oxide and this is not proving that Co³⁺ sites are the most active.

- page 7, it is mentioned that the HEO surface contains hydroxide or oxyhydroxide. How is this affecting the whole interpretation about an HEO activity based on missing more elements?

Minor comments:

- HEO synthesis: the sample is annealed in a flame and then cools down naturally. How fast is annealing?
- ECSA : The values of the capacitances given in inset of Fig. S14 look extremely small whereas Cdl are generally much larger at oxides. This is perhaps a question of numerical application. Please double check.

Reviewer #2 (Remarks to the Author):

I thank the authors for their reply which has clarified my primary concerns. From this I understand that the assumption made in the calculations is that the 5 nearest neighbors to the active site consist of exactly one of each of the elements Fe, Co, Cr, Mn and Ni. This is a reasonable way to reduce the number of computational structures, and as such I am satisfied that the computational setup is sensible. Nevertheless, this assumption should be expressed clearly in the manuscript, which is not currently the case, i.e., the sentence p4, 1116 should be modified (suggested changes in parentheses):

In this setting, each active site at the top-most layer is surrounded by the five nearest elements (neighbours) ($< 3.5 \text{ \AA}$) (A-E, Fig. 1a), and these five neighboring sites (are assumed to be occupied by exactly one atom of each of the elements Cr, Co, Mn, Ni and Fe in any arrangement, ...) can be occupied by Cr, Co, Mn, Ni, and Fe in any arrangements, which leads to 120 distinct permutations per active site.

Furthermore, regarding the reply to question 3:

It is shown that strain leads to the change in adsorption energy on Co. The unit cell is quite small, and on a real surface the strain from a Co with neighboring Fe might be relieved by the strain from a nearby Co with Mn neighbors. Did the authors investigate if the strain and resulting effects are the same in a larger unit cell with dissimilar active sites?

Our response:

We thank the reviewer for this suggestion to change the paragraph. We changed it accordingly.

In order to address the second portion of the reviewer's comment, we estimated the strain effects using larger unit cells and multiple active sites and the results show negligent effects. We observed that, despite changes in unit cell size, the strain inside the structure is generally unaffected. To demonstrate this, we chose the larger Mn_3O_4 bulk structure ($5.85 \times 5.85 \text{ \AA}$) as a surface structure rather than Co_3O_4 bulk ($5.75 \times 5.75 \text{ \AA}$) that is used in the original manuscript. The adjacent elements (Co, Ni, Mn, Fe, and Cr) are shown in the same orientation in **Figs. R1a** and **R1b** for the Co active site using different bulk unit cells, respectively. Although the unit cell in **Fig. R1b** is slightly larger (by 2%) than the original unit cell in **Fig. R1a**, the figures reveal that the lateral **Co-O** bond lengths are nearly identical.

Furthermore, we also tested the strain by increasing the unit cell by doubling the unit cell in both the x and y directions respectively. **Figs. R1c** and **R1d** show the unit cell and doubled unit cell structure after the geometry relaxation for the Co active site. In both cases, the lateral bond lengths between **Co-O** are nearly identical.

Fig. R1. Unit cell size effects on lateral strains developed in HEOs. O^* adsorption on the Co site using Co_3O_4 bulk (a) and Mn_3O_4 bulk (b) as the surface. O^* adsorption on Co site for single (c) and double (d) unit cells. The lateral strains remain constant regardless of the cell size.

Later, we investigated the strain effects using multiple cell sizes for active sites other than Co, such as Cr. We used original and doubled unit cells in the case of O^* adsorption on the Cr site for the HEO system (Fig. R2a, b). Despite doubling the cell size, the average lateral Cr-O bond distance stays also unchanged (Figs. R2c and R2d).

Fig. R2. Unit cell size effects on lateral strains developed in HEOs on Cr active site. It shows the average lateral Cr-O bond distances remain constant although cell size is different.

Reviewer #4 (Remarks to the Author):

This work about HEO catalysts is certainly interesting and seems solid. In the present form, authors convincingly show that mixing more elements improves the OER activity of the HEO but more care is requested to discuss differences between the OER activity of 4-5 elements HEOs. The current interpretation of this result remains fragile, and it needs to be said that the lack of operando XAS or operando XPS characterizations is highly regrettable. Operando structural characterization would also be desirable to check possible modifications of the catalyst and make sure that the surface is indeed as authors assume.

The revised paper needs further major revision.

Our response:

We appreciate the reviewer's thoughtful comments and suggestions. We agree that the inclusion of operando XAS or XPS characterizations would greatly support the interpretation of our activity results, but limited access to beamlines hindered us from conducting these experiments. Our current manuscript already encompasses numerous experiments and calculations due to the complexity of HEOs, and we plan to conduct a future operando study for our next publication. Nevertheless, we have strived to address the reviewer's comments and suggestions to the best of our abilities given our circumstances. Our responses and revisions can be found in this response letter and manuscript, and we believe that the manuscript, based on your suggestions, has been substantially improved.

Main comments:

1) About the dependence of OER activity as a function of number of elements within HEO.

A global trend seems obvious from the series of LSV. The OER activity increases by more elements. However, above 4 elements the differences between the CVs fall almost within the experimental reproducibility of the OER response. In particular, the overpotential difference is only 18 mV between CoFeNiCr and CoFeNiCrMn catalysts, which are the two systems at the heart of the discussion in this manuscript. Authors perfectly know that the CVs of independent samples of the same composition/structure may shift by several 10 mV. Therefore, authors should convince the reader that they are discussing real effects.

Recommendation: authors must provide additional electrochemical data to show that the dispersion of CVs is small compared to the 18mV discussed above. **They must show the CVs of several samples (in particular those mentioned above) prepared with the same composition. This will allow estimating an error bar for overpotentials, a parameter that is crucially missing.**

Response:

The reviewer is rightfully concerned with the OER activity difference among different spinel oxide materials, which is within the error bars range. Following the editor and reviewer's suggestion, we tested three to five samples for the same compositions of six different compositions (Co to CoFeNiCrMn) with the different iR corrections and summarized the error bars in the plot in **Supplementary Fig. 13 and Table 3**. These additional results support the trends we have

discussed. Incorporating Cr is effective to improve the activity as both CoFeNiCr and HEOs containing Cr show much higher activity than CoFeNi. Second, the inclusion of Mn helps to improve the stability as CoFeNiMn and HEO show higher stability than CoFeNi. So, we concluded that HEO is more active and stable than quaternary and quinary oxides.

Fig. 3 | Electrochemical response of various spinel oxides. a Linear sweep voltammetry (LSV) normalized to the geometric area with 90% iR correction in 1 M KOH electrolyte and **b** their Tafel plots. **c** Normalized LSV plots to the ECSA. **d** Overpotential plot as a function of the number of cations in oxides. **e** Chronopotentiometry (CP) curves of HEO anode (inset: CP comparison with other control groups). **(f-g)** Ex-situ XPS analysis. **f** the overpotential change and ex-situ XPS quantification results of the ratio of M³⁺ (M = Co, Fe, Ni, Cr, and Mn) to M over time. **g** The ratio of OH to (OH+O) over time.

Supplementary Fig. 13 | Linear sweep voltammetry curves based on the different iR corrections. a-f LSV curves for all multi-element spinel oxides (from binary to senary oxides) with different iR corrections. **g** Average resistance values for the different spinel oxides with error bars, and **h** their overpotential value to reach 10 mA cm^{-2} with error bars.

Supplementary Table 3 | Overpotential value to reach 10 mA cm⁻² in terms of different iR correction percentages.

Sample	Overpotential in terms of different iR correction percentage (Error bar included in Supplementary Fig. 13)				
	80%	85%	90%	95%	100%
Co	0.436	0.433	0.429	0.425	0.422
CoFe	0.402	0.398	0.395	0.392	0.388
CoFeNi	0.365	0.361	0.357	0.353	0.349
CoFeNiMn	0.352	0.348	0.345	0.341	0.338
CoFeNiCr	0.327	0.323	0.319	0.314	0.31
HEO	0.311	0.308	0.304	0.301	0.298

Revision:

We updated **Fig. 3** and included **Supplementary Fig. 13** and **Table 3** and the description of them in the manuscript.

(On page 10) “Among them, the HEO (CoFeNiCrMn) has the lowest overpotential of 304 mV at 10 mA cm⁻², in comparison to the other spinel oxides: CoFeNiCr (321 mV) < CoFeNiMn (331 mV) < CoFeNi (346 mV) < CoFe (385 mV) < Co (431 mV), and < Fe (522 mV) (**Supplementary Fig. 12**) with 90% iR correction. Here, we averaged the overpotentials with three to five samples. The different overpotential values in terms of different iR correction percentages are shown in **Supplementary Fig. 13** and **Supplementary Table 3.**”

2) Are mixed Co³⁺ sites the most OER active sites of the HEO?

The link between authors' statement and experimental characterizations remains fragile in absence of operando measurements. There is also a real problem when comparing DFT data with experiments. In Fig. 4c the predicted OER overpotential is 0.2 V larger for CoFeNiCr (0.49 V) than for CoFeNiCrMn (0.29 V) while these two catalysts present a rather similar OER activity (see above). This huge discrepancy here between DFT and experiments needs to be commented for it somewhat puts in question the whole reasoning.

Our response:

We thank the reviewer for pointing out this. We agree with the reviewer that the calculated overpotential trend for various spinel oxides shows a large difference from experimental observations. This disparity was possibly caused by the number of calculations we carried out. In

the initial manuscript, we simply removed the relevant element from the single (lowest overpotential) HEO structure to calculate the activity trend in **Fig. 4c**. Now, we have extended our calculation of the activity for other lower entropy oxides using the permutation method used in the manuscript for HEO systems. For 4 elements oxides, we used 4*4 permutation ($4P_4$) and for 3 elements oxides, we considered 3*3 permutation ($3P_3$) for Co active site. Updated results show that for 3 elements (Fe, Co, Ni), the global lowest overpotential is 0.47 V (**Supplementary Fig. 22**). Moreover, when we introduced Mn to the CoFeNi system, we obtained a new global minimum of 0.42 V overpotential (**Supplementary Fig. 23**) among 24 different combinations. Similarly, when Cr was added to the CoFeNi system, the predicted overpotential further decreased reaching its lowest global value of 0.37 V (**Supplementary Fig. 24**). Those additional results show a better-matched activity trend with experiment and the differences between two overpotentials in the earlier manuscript is now removed. The revised activity trend is shown in a new **Fig. 4c**. Nevertheless, it is impossible for DFT calculations to reproduce exactly the same overpotential as in experiments. We sincerely hope that the reviewer understands the limitations we faced and recognizes the substantial improvements made to the manuscript.

Supplementary Fig. 22. OER activity volcano plot as a 2D heat map of overpotentials for CoFeNi oxide system based on binding energy calculation of O* and OH* and scaled OOH* values for HEO systems. We considered 3*3 permutation ($3P_3$) for Co active site and found 6 different combinations. Among 6 combinations, we found the lowest overpotential of 0.47 V.

Supplementary Fig. 23. OER activity volcano plot as a 2D heat map of overpotentials for CoFeNiMn oxide system based on binding energy calculation of O* and OH* and scaled OOH* values for HEO systems. We considered 4*4 permutation (4P₄) for Co active sites and found 24 different combinations. Among 24 combinations, we predicted the lowest overpotential of 0.42 V for the following combinations.

Supplementary Fig. 24. OER activity volcano plot as a 2D heat map of overpotentials for CoFeNiCr oxide system based on binding energy calculation of O* and OH* and scaled OOH* values for HEO systems. We considered 4*4 permutation (4P₄) for Co active sites and found 24 different combinations. Among 24 combinations, we predicted the lowest overpotential of 0.37 V for the following combinations.

Fig. 4 | OER site-dependent activity of the equimolar HEO. **a** OER activity volcano plot as a 2D heat map of overpotentials based on binding energy calculation of O^* and OH^* , and scaled OOH^* values for HEO systems, grouped by the active site along with the pure spinel systems. **b** Active sites (Co, Fe, Cr) distribution as a function of OER overpotential for HEO systems. **c** Predicted activity trends for different spinel oxide systems with explicitly calculated OOH^* .

Revision:

We added **Supplementary Fig. S22-24** in the Supplementary Information and modified **Fig. 4c** in the manuscript.

(On page 14) “Later we calculated overpotential by considering permutation theory for lower entropy oxides as well (**Supplementary Figs. 22-24**). The calculated overpotential trends for these oxide systems are as follows: HEO (CoFeNiCrMn) (0.29 V) < CoFeNiCr (0.37 V) = Fe-NiOOH (0.37 V) < CoFeNiMn (0.42 V) < CoFeNi (0.47 V).”

3) About the role of microstrains on OER activity.

The data in Table S6 indeed show that strains increase by mixing more elements (btw, what this the reference crystal structure?). Nevertheless, these data do not prove that the micro strains are the cause for the enhanced OER activity.

Recommendation: To demonstrate their point, authors should show correlation OER activity – strains over the widest possible set of samples. They could measure the micro strains for all 4 to 5 elements HEOs they have synthesized and check whether they can establish correlation strain - activity.

Response:

Thanks for the great suggestion. First, our reference materials for HEO XRD analysis are $NiFe_2O_4$ (shown in the main figure) and $CoCr_2O_4$ as they are all spinel oxides. Also, we could not find any XRD reference that includes all five elements. The XRD peaks of HEOs are similar to $NiFe_2O_4$ and $CoCr_2O_4$ but some peaks are shifted due to the difference in the ionic radius of five elements,

leading to microstrain. As the reviewer recommended, we plotted the OER activity as a function of microstrain. In the plot, the x-axis represents the microstrain and the y-axis means the overpotential values. Here, at least three samples were analyzed to create the error bars in both microstrain and overpotentials. Indeed, it shows that when more elements are included, the microstrain increases and the onset potential reduces.

Supplementary Fig. S32 | Correlation plot of OER activity-microstrain. The x-axis represents the microstrain with the horizontal error bars and the y-axis means the overpotential values with the vertical error bars based on three samples for each condition.

Revision:

We included the microstrain vs. activity plot in **Supplementary Fig. S32**.

4) XAS:

-Panels (c-d) in Fig. S10 are confusing: These plots are supposed to be calibration plots and authors only show 2 data points corresponding to CoNiFeCr and CoNiFeMnCr catalysts and one straight line. This leaves the impression that the straight line is a “fit”, which would be meaningless with only 2 data points. Please show the data points corresponding to reference samples with their error bars. Give the value of slope and compare it with existing literature.

-In caption of Fig. S10, Fe₂O₃ is not Fe (III).

-Bulk oxidation states of elements should be compared with DFT values (this information is not found).

Response:

We acknowledge that fitting with only two data points is limited. Unfortunately, we could not measure XANES for other samples due to limited beam time access. To alleviate this issue, we have modified the plots so that the reader can easily compare the oxidation states of two samples based on the reference materials. As shown in **Supplementary Fig. S10**, we found that the

oxidation states for Co are 2.14 and 2.20 for CoFeNiCr and HEO, and those for Fe are 2.56 and 2.36, respectively. Here, we used the integral method to calculate those values based on the references, Environ. Sci. Technol. 2021, 55, 6042–6051 and Anal Bioanal Chem 413, 5395–5408 (2021). Generally, XAS displays the bulk oxidation states, not the surface states. We believe that this information can be employed to provide the basic material properties, while we are focusing on the surface oxidation states as the OER occurs at the surface-active sites.

In addition, we have revised the caption in **Supplementary Fig. 10**. “The reference oxides are Fe₂O₃ (Fe (III)) and Fe₃O₄ (Fe (II, III)).”

In order to compare the bulk oxidation states with the theoretical result, we calculated the average atomic Bader charge of Co for the bulk structure in HEO and CoFeNiCr system and tabulated as per their oxidation state below: (We randomly choose 10 structures and get their avg. Bader charge then extrapolated the approximate oxidation state as per their references). We find in the case of mixed oxide systems (HEO, CrCoFeNi) oxidation states are always between +2 and +3, but the oxidation state of Co in HEO is always higher than that in CoFeNiCr.

Systems	Co(OH) ₂	Co ₃ O ₄	CoOOH	HEO	CoFeNiCr
Avg. atomic Bader charge	1.17	1.28	1.31	~1.27	~1.29
Avg. oxidation states	+2	+2.67	+3	~+2.6	~+2.73

Similarly, for the Fe oxidation states are tabulated below. Here, the oxidation state of Fe in HEO is always lower than that in CoFeNiCr. This trend is well matched with the XANES result.

Systems	Fe(OH) ₂	Co ₃ O ₄	FeOOH	HEO	CoFeNiCr
Avg. atomic Bader charge	1.33	1.58	1.73	~1.67	~1.69
Avg. oxidation states	+2	+2.67	+3	~+2.81	~+2.89

Supplementary Fig. 10 | Normalized X-ray absorption near-edge structure (XANES) analysis for HEO with other reference oxides. a XANES spectra of the Co K-edge. The inset figure shows the expanded section from 7718 to 7725 eV. The references include CoO (Co (II)) and Co₃O₄ (Co (II, III)). **b** The calculated Co oxidation states from the XANES spectra. **c** Fe K-edge XANES spectra. The inset indicates the expanded region from 7123 to 7127 eV. The reference oxides are Fe₂O₃ (Fe (III)) and Fe₃O₄ (Fe (II, III)). **d** The calculated Fe oxidation states from the XANES spectra.

Revision:

We revised the XAS plots in **Supplementary Fig. S10** and the description in the main manuscript.

(On page 8) “Based on the X-ray absorption near-edge structure (XANES) profiles, the quantitative comparison of oxidation state from the Co K-edge spectra (**Supplementary Fig. 10a and 10b**) shows that the valence states of Co for both CoFeNiCr and HEO are located between Co²⁺ and Co²⁺/Co³⁺. In addition, the higher Co valence state appears with adding Mn cation to CoFeNiCr, which is consistent with the XPS result. Moreover, the Fe K-edge of CoFeNiCr and HEO is shown in **Supplementary Fig. 10c**, and the Fe oxidation state (**Supplementary Fig. 10d**) is in the lower region than the reference materials, Fe²⁺/Fe³⁺ and Fe³⁺ in which the relative Fe oxidation state of HEO is the lowest and tends to appear in a more divalent state. Thus, HEO exhibits relatively higher Co³⁺ than Co²⁺ and Fe²⁺ than Fe³⁺ throughout the particles, compared to those of the lower entropy oxide (CoFeNiCr).”

5) XRD:

-The reader wonders how XRD is experimentally combined with the flame annealing technique. Experimental details should be given.

-Fig. S4: the spectrum at 25°C measured is poorly commented. Was it measured after the sample has cooled down after annealing at 1000°C? Please specify. Bragg peaks must be indexed.

-Please show the Williamson-Hall plots and give error bars for the microstrains. It would be interesting to give the lattice parameters of the series of HEO.

Response:

The in-situ XRD is not combined with the flame synthesis method, and it is used to understand the phase changes of HEO. High-temperature XRD is operated by a slow heating process, which is different from the flame annealing process including fast heating and cooling. We could get insight into how the HEO crystal structure is developed at different temperatures using high-temperature XRD in which the spinel crystal structure appears after 400 °C. Our flame annealing is operated at ~1000 °C, so combined with the observation from high-temperature XRD, we could get the spinel structure for HEO. We have clarified this in the manuscript. “The in-situ temperature-dependent XRD results (**Supplementary Fig. 4**) show that the spinel structure starts to appear at 400 °C. In-situ temperature-dependent XRD is operated by a slow heating process, which is different from the flame annealing process including fast heating and cooling. We can still get insight into how the HEO crystal structure is developed at different temperatures. Thus, the spinel structure is prominent at the flame treatment temperature of 1000 °C, indicating that the sol-flame temperature is high enough to synthesize the spinel structure of HEO.”

For the second comment, the spectrum measured at 25 °C is the one cooled down after annealing at 1000 °C. We made a comment in the figure caption and also added the index.

For the final comment, we could obtain information on the size and strain broadening by considering peak width as a function of 2θ that Williamson-Hall proposed a method. By plotting $\text{FWHM} \times \cos \theta$ on the y-axis against $4\sin \theta$ on the x-axis, we can get the strain component from the slope and the particle size component from the y-intercept. We added this information in the figure caption and also, we measured at least three samples to make an error bar for each condition.

Supplementary Fig. 31 | Estimation of microstrain and crystallite size from XRD. **a** XRD result for several spinel oxides and **b** calculated crystallite size and microstrain for each oxide. Here, we used the Scherrer equation and Williamson-Hall plot to estimate the grain size and microstrain,

$$D = \frac{K\lambda}{\beta \cos\theta}$$

, where D is grain size, K = 0.9 (Scherrer constant), $\lambda = 0.15406$ nm, the wavelength of the x-ray sources (Cu K α), β = FWHM (in radians), and θ = peak position (in radians).

In the Williamson-Hall plot,

$$\beta \cos\theta = \varepsilon(4\sin\theta) + \frac{K\lambda}{D}$$

, where ε = microstrain. Finally, by plotting $\beta \cos\theta$ on the y-axis against $4\sin\theta$ on the x-axis, we can get the strain component from the slope and the particle size component from the y-intercept.

Revision:

We updated the microstrain plot by adding the error bars in **Supplementary Fig. S31** and corrected the explanation of this plot in the caption.

(On page 6) “The in-situ temperature-dependent XRD results (**Supplementary Fig. 4**) show that the spinel structure starts to appear at 400 °C. In-situ temperature-dependent XRD is operated by a slow heating process, which is different from the flame annealing process including fast heating and cooling. We can still get insight into how the HEO crystal structure is developed at different temperatures. Thus, the spinel structure is prominent at the flame treatment temperature of 1000 °C, indicating that the sol-flame temperature is high enough to synthesize the spinel structure of HEO.”

6) EIS:

The equivalent circuit given in Fig. S13b was originally proposed by Conway in Ref. 62 (with capacitance instead of CPE) and was also used in ref. 64 (see right scheme) but not in ref. 63. Conway used this scheme to discuss the mechanisms of HER accounting for the relaxation of surface coverage of an adsorbed intermediate (left figure): Rct is the usual charge transfer resistance, which accounts for the kinetics of the reaction; The parallel (Rp/Cp) circuit accounts for the relaxation of the surface coverage of an adsorbed intermediate in the HER mechanism and the CPE accounts for the dispersion of the Cdl. It seems legitimate to apply it for OER for reaction intermediates are also adsorbed during the reaction.

Authors' use the same circuit but interpret it quite different. In fact, R2 (analogue of Rct in left scheme) becomes the charge resistance inside bulk catalyst, whatever this means. The parallel R3/CPE circuit (analogue of Rp/Cp) becomes the charge transfer resistance between catalysts and electrolyte. This interpretation is not physically sound.

Authors must explain / justify their circuit and should not refer to Refs. 62-64 for their interpretation is different. Authors test their model. In particular, by discussing the two CPE, provided the factor “n” (not given) is close to 1 (otherwise the model is not suitable). The estimated Cdl could be compared with that used to estimate the ECSA.

Original scheme (Ref. 64) This submission: Scheme of Fig. S13

Response:

We really appreciate the reviewer raising this concern. The schematic of **Supplementary Fig. 14** is originally from the Gamry website as we used this circuit for our system. We agree that citing the circuit model in Ref 62-64 is not appropriate in our manuscript. We depicted our equivalent circuit model of an electrochemical cell with two electroactive interfaces. This circuit can be used to explain the charge transfer resistance between metal oxides and conductive substrates as well as series resistance and bulk charge transfer resistance. The interpretation of each resistance value is a series resistance to denote a resistance due to wires, contacts, and solutions (R1), charge transfer resistance inside bulk catalyst (R2), and charge transfer resistance at the interface between the catalyst and conductive substrate (R3). We referred to **ACS Appl. Energy Mater.** **2020**, **3**, **66–98** to interpret each resistance.

Supplementary Fig. 14 | Nyquist plots from EIS measurement for different multi-element spinel oxides and their equivalent circuit. The EIS results were measured at 1.54 V vs. RHE and fitted with a REAP2CPE equivalent circuit model in the Gamry instrument in which there were three types of resistances: resistance in the electrolyte solution (R1), charge transfer resistance inside bulk catalyst (R2), and charge transfer resistance at the interface between the catalyst and electrolyte (R3). Three resistance values are indicated in **Supplementary Table 4**.

Revision:

We changed the schematic of an equivalent circuit model in **Supplementary Fig. 14** with the appropriate description, removed ref 62-64, and added the correct reference.

7) XPS:

-The increase of the ratio $\text{Co}^{3+}/\text{Co}^{2+}$ after OER (Fig. S18) was to be expected because OER corresponds to strongly oxidizing conditions. This also happens at the surface of Co spinel oxide and this is not proving that Co^{3+} sites are the most active.

-page 7, it is mentioned that the HEO surface contains hydroxide or oxyhydroxide. How is this affecting the whole interpretation about an HEO activity based on missing more elements?

Response:

The intensity of the XPS peak is an indicator of the amount of constituent elements present on the surface. Changes in peak intensities can indicate atomic rearrangement or the formation of new phases during electrolysis. Here, the increased intensity of trivalent states of Co, Fe, and Ni suggests that these elements could be cations in (oxy)hydroxide species. Additionally, the higher peak intensity of OH in the O 1s region may result from (oxy)hydroxide formation. For future investigations, we hope to use ex-situ, post-situ, or possibly in-situ XAS or XPS to study the mechanism of atomic and structural arrangement during electrochemical measurements. This would help to identify structural and elemental changes that occur before, during, and after electrolysis. Many literatures reported the activity of different (oxy)hydroxide including Co, Fe, and Ni. Laia Francàs et. al. showed 570 mV and 500 mV for FeOOH and NiFeOOH at 2.5 mA cm⁻² (Fig. R3a) and even NiFe, CoFe, Co, and Ni hydroxides show much less activity than oxyhydroxides (Fig. R3b). Also, as shown in Fig. R3c and R3d, CoFe, CoCr, FeCr, and CoFeCr oxyhydroxides all display higher overpotentials than our HEO. Thus, despite the formation of (oxy)hydroxide during the measurement, the overpotential value remained lower than that of (Co-)Fe-NiOOH, indicating the contribution of HEO to activity.

Fig. R3 LSV data from **a** Nat Commun 10, 5208 (2019) **b** Nat Commun 11, 2522 (2020) **c and d** Chem. Mater. 2020, 32, 4303–4311.

Revision:

(On page 11) “Although it is well known that the surface oxyhydroxide is also a promising OER catalyst,⁶³ HEO still plays an important role as support contributing to the great stability. A previous report shows 570 mV and 500 mV of the overpotential for FeOOH and NiFeOOH at 2.5 mA cm⁻², respectively.⁶⁴ Even other oxyhydroxides including NiFe, CoFe, CoCr, FeCr, and CoFeCr all display higher overpotential than our HEO.^{65,66}”

Minor comments:

8) HEO synthesis: the sample is annealed in a flame and then cools down naturally. How fast is annealing?

Response:

Thanks for the comment. We put our samples inside the flame burner and the duration of the treatment is only 30 seconds. After the flame treatment, we just remove the sample from the flame (1000 °C to RT), letting it cool down naturally.

9) ECSA: The values of the capacitances given in inset of Fig. S14 look extremely small whereas Cdl are generally much larger at oxides. This is perhaps a question of numerical application. Please double check.

Response:

Thanks for pointing out this. We measured ECSA of at least three to five samples for each condition and recalculated the capacitance values with error bars in **Supplementary Fig. 15**.

Supplementary Fig. 15 | Electrochemical surface area (ECSA) analysis. **a** The difference between anodic and cathodic current density as a function of scan rate, and **b** the double layer capacitance values for the different spinel oxides.

Revision:

(On page 10) “In addition, based on the measured electrochemical surface area (ECSA) with their double-layer capacitance (**Supplementary Fig. 15**), **Fig. 3c** shows the polarization curves of different spinel oxides normalized by ECSA. The trend shows the same as those with the geometric area, but the normalized current density was decreased due to a much higher electrochemical surface area.”

Second Revised Manuscript NatComm_379092_0

The authors addressed most of queries and made substantial improvement. A few key points yet deserve additional care and work. The revised paper needs further revision.

1) OER activity vs. composition:

Authors did not address the initial queries about the *reproducibility* of the EC behavior of samples with a given composition but issued from different batches. It was asked "... showing the CVs of several samples (in particular those mentioned above) prepared with the same composition. This will allow estimating an error bar for overpotentials, a parameter that is crucially missing". This remark was especially referring to the data in Fig. 3d.

- In this context showing LSV with different iR drop correction is pointless (SI, Fig. 13).

- To determine the iR drop correction authors had better looking at Tafel plots to adjust "R". When a straight line is obtained after iR correction, this gives a good approximation of the series resistance. In this regard, the curvature of Tafel plots in Fig. 3 suggests the correction was not optimum.

- The physical origin of error bars in Fig. 13 is not explained and is, anyway, not the one expected from the initial report.

2) Normalization of OER current by ECSA:

- The values of the capacitances given in of Fig. S15 are now in the F/cm² range. Please add the value of the FTO electrode for comparison. These values probably mean an ECSA that is roughly 10-20 times greater than the FTO surface. Can authors estimate whether this is consistent with the morphology of samples?

- Is not it strange that the shape of LSV plots evolves by normalizing the current by the ECSA in Fig. 3a and 3c? Look in particular the plot for Co. The green plot is also strange in 3c. Could this be related to a lack of iR correction?

3) EIS:

Authors did not address previous comments and request. Relying on the work **ACS Appl. Energy Mat. 2020** is very short. *Honestly, this part should be removed for authors do not make use of it to yield additional insights.* In addition, the analysis of these data raises mainly problems.

We draw attention of the fact that Model (C) in Table 2 (see Fig. 1A) of this reference is justified when the EIS spectra present well separated semi circles. This is NOT the case of the EIS data presented by authors (Fig. 1B). Therefore, a simpler circuit like Model (A) of the same reference must be tested. In the present form there is no justification for the complex equivalent circuit used by authors. In addition, it may be strongly suspected that the fit gives strongly correlated fitting parameters because of a too large number of parameters with respect to the amount of data.

Authors should show test models with the least possible number of parameters and demonstrate that adding one element is decisive. A comparison between experimental and calculated EIS with different models is requested. Moreover, the final Table 4 (SI) must give all element parameters of the equivalent circuit. In particular, the exponent parameter of the CPE which must be in the range 1-0.8 and the capacitances. The interface capacitance should be compared with the one derived from CVs used to determine the ECSA.

4) The correlation strain – activity is nice. However, one should remain cautious because it not a clear cut proof that strains are at the origin of the enhanced OER activity. Other concomitant / correlated effects are also at stake, such as the surface composition. This should be clearly outlined in the conclusion.

Third Revised Manuscript NatComm_379092_0

The authors addressed most of the queries and made substantial improvements. A few key points yet deserve additional care and work. The revised paper needs further revision.

Our response:

We deeply value the feedback and recommendations provided by the reviewer to enhance the quality of our work. We have made diligent efforts to respond to and incorporate all the comments provided. As a result of the comprehensive review process conducted by the reviewer, our manuscript has undergone significant improvements. Our response to reviewer comments will be in blue and the revised parts in red color, which are highlighted in yellow color in the revised manuscript.

1) OER activity vs. composition:

Authors did not address the initial queries about the reproducibility of the EC behavior of samples with a given composition but issued from different batches. It was asked "... showing the CVs of several samples (in particular those mentioned above) prepared with the same composition. This will allow estimating an error bar for overpotentials, a parameter that is crucially missing". This remark was especially referring to the data in Fig. 3d.

Our response:

We have incorporated the reproducibility plot for the electrochemical behavior in **Supplementary Fig. 13h**, which is shown below. In this plot, we have included error bars by measuring at least three samples for each composition. We acknowledge that it may appear confusing as it encompasses different iR correction percentages. However, in the main manuscript, we employed 90% iR compensation, and the error ranges of the overpotential for different compositions can be identified. Additionally, in response to the reviewer's request, we have plotted the LSV curves for HEO (four samples) and CoFeNiCr (three samples) to assess reproducibility (**Fig. R1**). We concur with the reviewer that there can be variations in current densities among samples due to factors such as substrate resistance and coverage. We will include a description regarding the reproducibility of different compositions and the potential for performance variation.

Fig. R1 LSV curves of two different compositions (HEO vs. CoFeNiCr) to see the sample variation.

Figure S13h Overpotential values with different iR correction percentages for different compositions. The error bars are evaluated from three or more samples of the same chemical composition.

Revision:

(On page 10) “To assess the reproducibility of the same composition, we conducted electrochemical measurements for at least three samples of the same composition to calculate the error bars. The slight variations in current densities among samples are attributed to experimental factors, such as substrate resistance, surface coverage, and other relevant variables.

- In this context showing LSV with different iR drop correction is pointless (SI, Fig. 13). To determine the iR drop correction authors had better looking at Tafel plots to adjust “R”. When a straight line is obtained after iR correction, this gives a good approximation of the series resistance. In this regard, the curvature of Tafel plots in Fig. 3 suggests the correction was not optimum.

Our response:

We added the LSV curves in terms of different percentages of iR compensation, in response to the editor’s request in the previous round, “Our editorial team also share ..., thus request the reporting of the resistance of your system with error bar as well as considering other levels of iR compensation.” We agreed with this as data can be easily compared to other literature since others may use different percentages of iR compensation (from 80 to 95%). We have also replotted the Tafel plots for better quality.

Fig. 3 | Electrochemical response of various spinel oxides. a Linear sweep voltammetry (LSV) normalized to the geometric area with 90% iR correction in 1 M KOH electrolyte and **b** their Tafel plots. **c** Normalized LSV plots to the ECSA. **d** Overpotential plot as a function of the number of cations in oxides...

- The physical origin of error bars in Fig. 13 is not explained and is, anyway, not the one expected from the initial report.

Our response:

We have taken into consideration the reviewer's comment and have now included a description in the figure caption to provide clear and concise information regarding the origin of the error bars.

Revision:

(Supplementary Information) Supplementary Fig. 13 | Linear sweep voltammetry curves based on the different iR corrections. a-f LSV curves for all multi-element spinel oxides (from binary to senary oxides) with different iR corrections. **g** Average resistance values for the different spinel oxides with error bars, and **h** overpotential values with different iR correction percentages for different compositions. The error bars are evaluated from three or more samples of the same chemical composition.

2) Normalization of OER current by ECSA:

- The values of the capacitances given in Fig. S15 are now in the F/cm² range. Please add the value of the FTO electrode for comparison. These values probably mean an ECSA that is roughly 10-20 times greater than the FTO surface. Can authors estimate whether this is consistent with the morphology of samples? -Is not it strange that the shape of LSV plots evolves by normalizing the current by the ECSA in Fig. 3a and 3c? Look in particular the plot for Co. The green plot is also strange in 3c. Could this be related to a lack of iR correction?

Our response:

Thanks for pointing out this. We have incorporated the double-layer capacitance for the bare FTO substrate and corrected the unit from F/cm^2 to mF to easily calculated ECSA and Roughness factor. While it is true that roughness/morphology can be correlated with activity, it is important to note that it is not the sole determining factor. This is evident from the observation that CoFeNi and CoFeNiMn, despite having higher roughness factors, exhibit lower activities compared to CoFeNiCr and HEO. We believe that the enhanced activity is a result of the synergy among various factors, and multiple parameters contribute to the overall performance.

In detail, in terms of the double-layer capacitances for different compositions, the ECSA values are several hundred times higher than that of the FTO substrate. For the calculation of the current density normalized by ECSA, we used the equation, $ECSA = C_{dl}/C_s$, where C_s is the specific capacitance typically ranging from $20\text{--}60 \mu F \cdot cm^{-2}$ in alkaline solutions. [*J. Am. Chem. Soc.* 2013, 135, 16977–16987, *J. Am. Chem. Soc.* 2015, 137, 4347–4357] Here, we took $40 \mu F \cdot cm^{-2}$ as the C_s value. The roughness factor can also be determined by dividing the electrochemical surface area (ECSA) by the geometric surface area of the electrode. For the respective materials FTO, Co, CoFe, CoFeNi, CoFeNiMn, CoFeNiCr, and HEO, the resulting roughness factor values are 0.12, 15.20, 20.39, 41.63, 46.92, 19.96, and 23.28, respectively.

The reviewer made a note about the strange shape of the LSV curves, which was attributed to the different ranges being plotted. The original shape of the current curves resembles that shown in Figure 3a, as they were all divided by the same geometric area. However, when dividing the curves by ECSA and plotting them in different ranges, they exhibit variations in appearance. Now we have modified the scale of the axis.

We have added that information on ECSA and roughness factor to the caption in **Supplementary Fig. 15**. Finally, we normalized the current density by ECSA and revised the plot in **Fig. 3c**, accordingly.

Supplementary Fig. 15 | Electrochemical surface area (ECSA) analysis. **a** The difference between anodic and cathodic current as a function of scan rate, and **b** the double layer capacitance values for the different spinel oxides. At least three samples were used for the measurement. **c** ECSA calculated by $ECSA = C_{dl}/C_s$, where C_s is the specific capacitance, which is usually used to be about $20\text{--}60 \mu\text{F}\cdot\text{cm}^{-2}$ in alkaline solutions. We took $40 \mu\text{F}\cdot\text{cm}^{-2}$ as the C_s value. **d** Roughness factor of the electrodes calculated by dividing ECSA by the geometric area.

Fig. 3 | Electrochemical response of various spinel oxides. **a** Linear sweep voltammetry (LSV) normalized to the geometric area with 90% iR correction in 1 M KOH electrolyte and **b** their Tafel plots. **c** Normalized LSV plots to the ECSA...

Revision:

(On page 10) “Furthermore, utilizing the measured ECSA along with the corresponding double-layer capacitance, Fig. 3c illustrates the polarization curves of various spinel oxides normalized by ECSA. A comprehensive description of the ECSA can be found in Supplementary Fig. 15. The roughness factor was also determined by dividing the ECSA by the geometric surface area of the electrode. For the respective materials FTO, Co, CoFe, CoFeNi, CoFeNiMn, CoFeNiCr, and HEO, the resulting roughness factor values are 0.1, 15.2, 20.4, 41.6, 46.9, 20.0, and 23.3, respectively. The LSV curve trend after normalizing by ECSA depicted in Fig. 3c shows the same as those with the geometric area. However, the normalized current density appears to decrease due to the significantly higher electrochemical surface area.”

3) EIS:

Authors did not address previous comments and request. Relying on the work ACS Appl. Energy Mat. 2020 is very short. Honestly, this part should be removed for authors do not make use of it to yield additional insights. In addition, the analysis of these data raises mainly problems. We draw attention of the fact that Model (C) in Table 2 (see Fig. 1A) of this reference is justified when the EIS spectra present well separated semi circles. This is NOT the case of the EIS data presented by authors (Fig. 1B). Therefore, a simpler circuit like Model (A) of the same reference must be tested. In the present form there is no justification for the complex equivalent circuit used by authors. In addition, it may be strongly suspected that the fit gives strongly correlated fitting parameters because of a too large number of parameters with respect to the amount of data. Authors should

show test models with the least possible number of parameters and demonstrate that adding one element is decisive. A comparison between experimental and calculated EIS with different models is requested. Moreover, the final Table 4 (SI) must give all element parameters of the equivalent circuit. In particular, the exponent parameter of the CPE which must be in the range 1-0.8 and the capacitances. The interface capacitance should be compared with the one derived from CVs used to determine the ECSA.

Our response:

We appreciate and understand the reviewer's concern regarding EIS and generally OER catalyst research papers use Model A, B, or C (in *ACS Appl. Energy Mater.* 2020, 3, 66–98, attached in **Fig. R4**) as EIS circuit models. First, as suggested by the reviewer, we conducted additional measurements of EIS by fitting with Model A. Our conclusion remains consistent, confirming that HEO demonstrates the lowest charge transfer resistance at the catalyst-electrolyte interface. We also have described the details in **Supplementary Fig. 14**.

The details are here; Model A represents a simple Randles circuit model, which describes a single solid/electrolyte interface with both Faradaic and non-Faradaic current flows. Model C, on the other hand, presents an equivalent circuit of a system with two interfaces exhibiting different kinetics. In this case, additional elements (R2 and Q2) are connected in parallel to each other and in series with R_s and R1, which are part of the existing elements of the simple Randles model. This occurs when the substrate electrode modified with an electrocatalyst is not fully covered, and the substrate itself is also active for the electrocatalysis (e.g., carbon fiber paper, glassy carbon, F-doped SnO₂, In-doped SnO₂, Ti mesh, Ni foam, etc.).

To address the reviewer's request, we attempted fitting and simulation using both Model A and Model C (Now, Model **c** and Model **f** in **Supplementary Fig. 14**, respectively). As the reviewer mentioned, the result may not look like two combined semicircles. First, when the result is fitted by using Model **c** simple Randles circuit (one semicircle) (**Supplementary Fig. 14 c-e**), the fitted curves in the lower impedance region do not fit exactly with the raw data (**Supplementary Fig. 14e**), but overall it looks good. When we employ Model **f** (**Supplementary Fig. 14 f-h**), the results appear to fit well. That is why we previously used Model **f** for our results. At this time Q1 and Q2 values are 1.7e-4 F and 1.2e-4 F. According to the literature, when Q1 > Q2, Model **f** resembles a Randles circuit (Model **c**) and exhibits a single arc in the Nyquist plot defined by the total resistance of R1+R2. If Q1 and Q2 are similar in magnitude, the shape of the arc may appear warped or misshapen due to overlapping frequency responses from each capacitor. In our case, Q1 > Q2, but they are still comparable, so both models could potentially be used.

Thus, to simplify our explanation, and as the reviewer suggested, we re-measured EIS and switched the fitting model to Model **c**. It is important to note that the EIS was measured at 1.54 V vs. RHE, which is not a non-Faradaic region for the samples, so the interface capacitance values cannot be directly compared to those derived from ECSA. Again, our conclusion is same as the previous one, confirming that HEO exhibits the lowest charge transfer resistance at the interface between the catalyst and electrolyte.

Fig. R4 Different EIS circuit models from *ACS Appl. Energy Mater.* 2020, 3, 66–98.

Revision:

(Supplementary Information)

Supplementary Fig. 14 | Nyquist plots from EIS measurement for different multi-element spinel oxides and their equivalent circuit. a The EIS results for six different compositions and **b** the zoomed-in plots. The EIS results were obtained at 1.54 V vs. RHE and fitted with an equivalent circuit model in **c**. This model incorporated two types of resistances: R_s , representing any resistances associated with solution resistance and other electrical contacts, and R_1 , representing the charge transfer resistance at the interface between the catalyst and electrolyte. Two resistance values and double-layer capacitance are indicated in **Supplementary Table 4. d, e** EIS results of CoFeNi fitted by model **c**. **g, h** EIS results of CoFeNi fitted by model **f**.

Model **c** represents a simplified Randles circuit model, which characterizes a single interface between a solid and electrolyte. On the other hand, Model **f** presents an equivalent circuit that encompasses a system with two interfaces exhibiting different kinetics. In Model **f**, additional elements (R_2 and Q_2) are added into the simple Randles model. This configuration is employed when the substrate electrode modified with an electrocatalyst is not fully covered, and the substrate itself also participates in the electrocatalysis process. Examples of such substrates include carbon fiber paper, glassy carbon, F-doped SnO_2 , In-doped SnO_2 , Ti mesh, Ni foam, and others.

Also, when $Q_1 > Q_2$, Model **f** resembles a Randles circuit and displays a single arc in the Nyquist plot, determined by the total resistance of R_1+R_2 . If Q_1 and Q_2 are comparable in magnitude, the shape of the arc may appear distorted or warped due to overlapping frequency responses from each capacitor. In our specific case, although $Q_1 > Q_2$, the magnitudes are still comparable, suggesting that both models could potentially be applicable.

4) The correlation strain – activity is nice. However, one should remain cautious because it not a clear cut proof that strains are at the origin of the enhanced OER activity. Other concomitant / correlated effects are also at stake, such as the surface composition. This should be clearly outlined in the conclusion.

Our response:

We greatly appreciate the comments and suggestions provided by the reviewer. In response, we have added references and also have taken care to acknowledge the presence of other factors and their influence in the discussion section of our study.

Revision:

(On page 15) “Indeed, microstrain in material structures is recognized as a significant factor impacting OER activity.^{70,71} Nevertheless, we recognize that the OER performance is also influenced by other factors, such as surface composition. Therefore, it is vital to consider that microstrain is one of these factors affecting the OER activity.”

REVIEWERS' COMMENTS

Reviewer #4 (Remarks to the Author):

Dear Authors,

this reviewer is thankful for your revision. The electrochemical data appear robust and the work is overall of excellent quality.

We could not agree more with you that the catalytic activity is not just a matter of ECSA. This would be of no interest. This is why it is necessary, in a first step, to normalize the measured current with respect to the ECSA. By doing this you highlight that the intrinsic activity of the catalysts is involving other effects as you mention.

The paper may be accepted as is after having double checked the roughness factor of the FTO electrode. It cannot be < 1 . This is not physical. Double checking the data analysis is mandatory. It might simply be a question of units. Please, modify text and figures accordingly.

Optional revision and comment about EIS: The analysis with the simplest model is now acceptable though the description of data could have been more complete. You could have given the exponent value of the CPE and calculate the equivalent capacitance (provided the exponent is > 0.8). Such measurements in the pre-OER region would give you an additional insight into the ECSA. Regarding the use of model f, the addition of parameters is indeed expected to improve the quality of the fit. However, parameters values become probably correlated and more difficult to interpret.

Final Revised Manuscript NatComm NCOMMS-22-29265C

Reviewer #4 (Remarks to the Author):

Dear Authors,

This reviewer is thankful for your revision. The electrochemical data appear robust and the work is overall of excellent quality.

We could not agree more with you that the catalytic activity is not just a matter of ECSA. This would be of no interest. This is why it is necessary, in the first step, to normalize the measured current with respect to the ECSA. By doing this you highlight that the intrinsic activity of the catalysts is involving other effects as you mention.

The paper may be accepted as is after having double-checked the roughness factor of the FTO electrode. It cannot be < 1 . This is not physical. Double-checking the data analysis is mandatory. It might simply be a question of units. Please, modify the text and figures accordingly.

Optional revision and comment about EIS: The analysis with the simplest model is now acceptable though the description of data could have been more complete. You could have given the exponent value of the CPE and calculated the equivalent capacitance (provided the exponent is > 0.8). Such measurements in the pre-OER region would give you additional insight into the ECSA. Regarding the use of model f, the addition of parameters is indeed expected to improve the quality of the fit. However, parameter values become probably correlated and more difficult to interpret.

Our response:

We greatly appreciate the valuable suggestions offered by the reviewer, which leads to substantial improvements in our manuscript. The roughness factor of the bare FTO substrate was recalculated as there was a calculation error due to the confusion of the unit. Now we thoroughly double-checked the value and made the necessary revisions accordingly in this letter, Supplementary Information, and manuscript file.

As suggested by the reviewer, we used the simple Randles circuit model (model c in Supplementary Fig. 14) for the EIS analysis instead of the complex model (model f). However, we have still included the details of both circuit models in the caption, indicating that both models could be valid for the analysis.

Supplementary Fig. 15 | Electrochemical surface area (ECSA) analysis. **a** The difference between anodic and cathodic current as a function of scan rate, and **b** the double layer capacitance values for the different spinel oxides. At least three samples were used for the measurement. **c** ECSA calculated by $ECSA = C_{dl}/C_s$, where C_s is the specific capacitance, which is usually used to be about 20–60 $\mu\text{F}\cdot\text{cm}^{-2}$ in alkaline solutions. We took 40 $\mu\text{F}\cdot\text{cm}^{-2}$ as the C_s value. **d** Roughness factor of the electrodes calculated by dividing ECSA by the geometric area.

Revision:

(On page 8) “For the respective materials FTO, Co, CoFe, CoFeNi, CoFeNiMn, CoFeNiCr, and HEO, the resulting roughness factor values are 1.2, 15.2, 20.4, 41.6, 46.9, 20.0, and 23.3, respectively.”